# Undenatured type II collagen protects against collagen-induced arthritis by restoring gut-joint homeostasis and immunity

Piaopiao Pan[1], Yilin Wang[2], Mukanthu H. Nyirenda[3,5], Zainulabedin Saiyed[4], Elnaz Karimian Azari[4], Amy Sunderman[4], Simon Milling[3], Margaret M. Harnett[3] & Miguel Pineda [1] ✉

Oral administration of harmless antigens can induce suppression of reactive immune responses, a process that capitalises on the ability of the gastrointestinal tract to tolerate exposure to food and commensal microbiome without triggering inflammatory responses. Repeating exposure to type II collagen induces oral tolerance and inhibits induction of arthritis, a chronic inflammatory joint condition. Although some mechanisms underlying oral tolerance are described, how dysregulation of gut immune networks impacts on inflammation of distant tissues like the joints is unclear. We used undenatured type II collagen in a prophylactic regime -7.33 mg/kg three times/week- to describe the mechanisms associated with protective oral immune-therapy (OIT) in gut and joint during experimental Collagen-Induced Arthritis (CIA). OIT reduced disease incidence to 50%, with reduced expression of IL-17 and IL-22 in the joints of asymptomatic mice. Moreover, whilst the gut tissue of arthritic mice shows substantial damage and activation of tissue-specific immune networks, oral administration of undenatured type II collagen protects against gut pathology in all mice, symptomatic and asymptomatic, rewiring IL-17/IL-22 networks. Furthermore, gut fucosylation and microbiome composition were also modulated. These results corroborate the relevance of the gut-joint axis in arthritis, showing novel regulatory mechanisms linked to therapeutic OIT in joint disease.

Rheumatoid arthritis (RA) is a chronic, autoimmune inflammatory condition affecting primarily the synovial joints, although inflammation can affect other tissues like the skin, eye, heart, lung, kidney and gastrointestinal systems[1,2]. Despite advancements in immunosuppressive treatments, approximately 20–30% of patients still do not respond to them, and a clear increase of incidence rates in the global ageing population is described[3]. Thus, new therapeutic interventions are needed, but the aetiology of RA is not yet completely understood, and most disease triggers are still unidentified. Although genetic factors are involved, it is known that environmental effects can play a key role in disease initiation and progression. For example, smoking can induce post-translational modifications leading to major histocompatibility complex (MHC) presentation of modified self-

proteins to T cells. As a result, autoimmune responses are initiated, including the production of self-reactive antibodies against immunoglobulin G (Rheumatoid Factor), antibodies to citrullinated protein antigens (ACPAs)[4,5] or the cartilage component collagen type II (C-II)[6]. Interestingly, and contrary to the heterogenous pool of antigens recognised by ACPAs or Rheumatoid Factor, type II collagen, being a single molecule, constitutes a good target for clinical interventions based on oral tolerance mechanisms. Experimentally, responses against collagen can be studied in vivo using the collagen-induced arthritis (CIA) model[7], in which intradermal injections of type II collagen and Freud's adjuvant break tolerance and initiate inflammatory responses in the joint. Data generated with the CIA model have shown that repeated oral collagen administration stimulates systemic

[1]Centre for the Cellular Microenvironment, School of Molecular Biology, University of Glasgow, Glasgow, UK. [2]Department of Bacteriology and Immunology, Beijing Chest Hospital, Capital Medical University/Beijing Tuberculosis & Thoracic Tumor Research Institute, Beijing, China. [3]Institute of Infection and Immunity, University of Glasgow, Glasgow, UK. [4]Research and Development, Lonza Greenwood LLC, North Emerald Road, Greenwood, SC, USA. [5]Present address: Institute of Immunology and Immunotherapy, College of Medical and Dental Sciences, University of Birmingham, Birmingham, UK.
✉e-mail: miguel.pineda@glasgow.ac.uk

tolerance[8–10]. Various underpinning mechanisms have been described including expansion of tolerogenic dendritic cells in the gut-associated lymphoid tissue (GALT), with subsequent induction of antigen-specific Tregs, IL-10 and TGFβ responses[9,11,12] and suppression of inflammatory IL-17 pathways[13]. These positive results in pre-clinical studies encouraged further development of collagen-based formulations. In this study, we used undenatured type II collagen, a form of collagen isolated from the cartilage of chicken sternum that preserves the physiological structure of collagen fibres[14]. Unlike the production of denatured collagen molecules, the undenatured type II collagen manufacturing process maintains the intact glycosylation and protein tertiary structure[15] that is required to shape the protein's characteristic triple helix[16]. The preservation of this structure is key because it determines the three-dimensional epitopes presented to immune cells in the Peyer's Patches of the gut[17]. Furthermore, by resisting the hydrolytic action of human gastric fluid[15], the preservation of triple helix domains allows a more efficient presentation of epitopes and subsequent regulatory responses. Supporting these findings, some studies indicated that undenatured type II collagen reduces pain and joint stiffness in osteoarthritis, both in experimental models and patients[18–20]. Mechanistically, oral collagen administration promotes regulatory T cell function in the gut tissue, increasing IL-4, IL-10 and TGFβ levels[20], whilst down-regulating inflammatory cytokines (IL-2, TNF, IL-6 and IL-1) and metabolites, like β-hydroxybutyrate[21–23]. Thus, undenatured type II collagen has also shown efficacy in ameliorating pain in joint disease[15], although the precise immunological basis of protection remains unclear. Perhaps surprisingly, less attention has been given to the structural aspects of the gastrointestinal regions where the described effector mechanisms take place. Recent research indicates that maintenance of the local gut tissue architecture and microenvironmental cues are critical for correct immunological discrimination and homoeostatic resolution of inflammation. Loss of mucosal barrier function in the gut has been implicated in the aetiology of arthritis[24,25] suggesting that restoration of the gut barrier integrity could be exploited therapeutically. Moreover, it has also been postulated that RA could initiate at mucosal sites before transitioning to more distant synovial joints[26]. Thus, a better understanding of the microenvironmental changes in gut mucosal sites in health and disease will allow us to understand and ultimately utilise, induction of oral tolerance.

In this study, we use the murine model of CIA to further characterise (i) immune mechanisms activated upon administration of prophylactic doses of undenatured type II collagen in mice, and (ii) functionally relevant structural changes of the gut mucosal sites associated with undenatured type II collagen protection against CIA. Using this system as a model of oral immune therapy (OIT), we show that mice treated with undenatured type II collagen present a significant reduction in disease incidence, which is associated with the regulation of IL-17 and IL-22 cytokines in the joint and rewiring of cellular and cytokine networks in the gut. Specifically, OIT was associated with the protection of gut villi and crypt structure and the maintenance of local levels of fucosylation. In describing the local gut responses by which OIT acts to suppress inflammatory responses, these findings increase our fundamental understanding of the role(s) of the gut in autoimmunity and will support the development of alternative areas for clinical intervention and prevention in RA.

## Results
### Undenatured type II collagen protects against experimental arthritis
We chose the CIA model to study the effects of oral administration of undenatured collagen in arthritis progression. We defined three experimental groups: (i) non-treated animals (Naïve), (ii) mice undergoing experimental arthritis (CIA) and (iii) mice undergoing arthritis with prophylactic undenatured type II collagen oral immunotherapy (OIT). OIT started 2 weeks prior to CIA induction, and oral gavage of undenatured type II collagen was applied 3 times per week. All mice in the CIA group developed symptoms. Monitoring of disease progression showed that OIT suppressed joint inflammation as evidenced by the significant reduction in

disease scores and incidence (Fig. 1a), showing that ~50% of the mice did not show any clinical symptoms. To evaluate whether OIT impacted predominantly on disease incidence, or it was influencing both incidence and severity of disease, we divided OIT mice into symptomatic and asymptomatic groups to measure paw swelling and relative weight change (Fig. 1b). OIT symptomatic mice were not significantly different to CIA controls, in terms of paw swelling and weight loss, whilst OIT asymptomatic mice showed similar values to healthy naïve controls. Values for individual mice are represented in Supplementary Fig. 1.

Corroborating the clinical results, histopathological analysis of CIA joints showed high numbers of infiltrating immune cells, causing bone damage and severe loss of cartilage that was absent in those OIT mice that were asymptomatic (Fig. 1c). Quantification of these disease indicators showed that asymptomatic OIT mice were completely protected against cartilage and bone damage in all cases, although they presented low levels of cell infiltration and pannus formation in some cases (Fig. 1d). Symptomatic OIT mice showed higher percentages of mice with no pathological signs for cell infiltration, cartilage damage and bone damage, although the results were not statistically significant and overall, scores were similar to those observed in the CIA control group (Fig. 1d).

### OIT inhibits systemic inflammatory and cellular pathways
Disease scores and histology demonstrated the therapeutic effect of undenatured type II collagen, protecting around 50% of the CIA mice from developing symptomatic arthritis. To gain insight into the protective mechanisms, we first evaluated the effect on pathogenic humoral responses as determined by the levels of IgG1 and IgG2a anti-CII antibodies[27]. As expected, all mice undergoing CIA had increased levels of anti-collagen antibodies in serum, with OIT mice exhibiting slightly lower anti-CII IgG1 but similar IgG2a levels compared to CIA (Fig. 2a). Nevertheless, both symptomatic and asymptomatic mice generated high levels of anti-CII antibodies, suggesting that the inhibition of humoral responses was not a pivotal factor in preventing arthritis development in our model.

Hence, we next assessed cellular immunity, working under the hypothesis that cellular immune responses were rewired in protected OIT mice. First, we counted the total number of cells in joint-draining lymph nodes (DLN), i.e., axillary, brachial and popliteal. As expected, CIA mice had a significantly higher number of cells compared to naïve, whereas the numbers in naïve and asymptomatic OIT were not significantly different and symptomatic OIT mice resembled CIA controls (Fig. 2b). Further analysis of distinct immune cell populations by flow cytometry showed that this trend was conserved in B and T cells, including CD4+ and CD8+ subsets, as all these lymphocyte groups in asymptomatic OIT were not significantly different from the levels found in naïve mice (Fig. 2c).

Since regulatory T cells (Tregs) have been extensively implicated in the establishment of tolerance, we evaluated the levels of CD3+CD25+FoxP3+ Tregs in the DLNs of asymptomatic OIT mice compared to those of the naïve and CIA groups (Supplementary Fig. 2a). We also evaluated CD39 and CD73 expression as functional indicators of their effector suppressive capacity, since these markers are regarded as immunological switches that shift ATP-driven pro-inflammatory immune cell activity towards an anti-inflammatory state mediated by adenosine[28] (Supplementary Fig. 2b, c). We did not observe any expansion of Tregs in asymptomatic OIT mice at the full day (day 33), suggesting that (systemic) Treg-mediated mechanisms were not responsible for the observed undenatured type II collage n protection against CIA during this late stage. Although we cannot rule out that Tregs play a role in protection during pre-clinical disease stages, we hypothesised that protection in CIA-OIT mice was associated with a reduction in pro-inflammatory cytokines. Specifically, we investigated the expression of IL-17, a highly pathogenic factor in CIA[29], and IL-22, a cytokine we reported to promote the development of joint disease[30–33]. To confirm the pathogenic role of IL-17 and IL-22 in the CIA joint, we first collated mice from all groups to directly compare their clinical scores with IL-17 and IL-22 expression in CD4+ T cells (Fig. 3a) and B cells (Fig. 3b), analysed by flow cytometry. Results indicate that levels of IL-17+ and IL-22+ CD4 T cells significantly

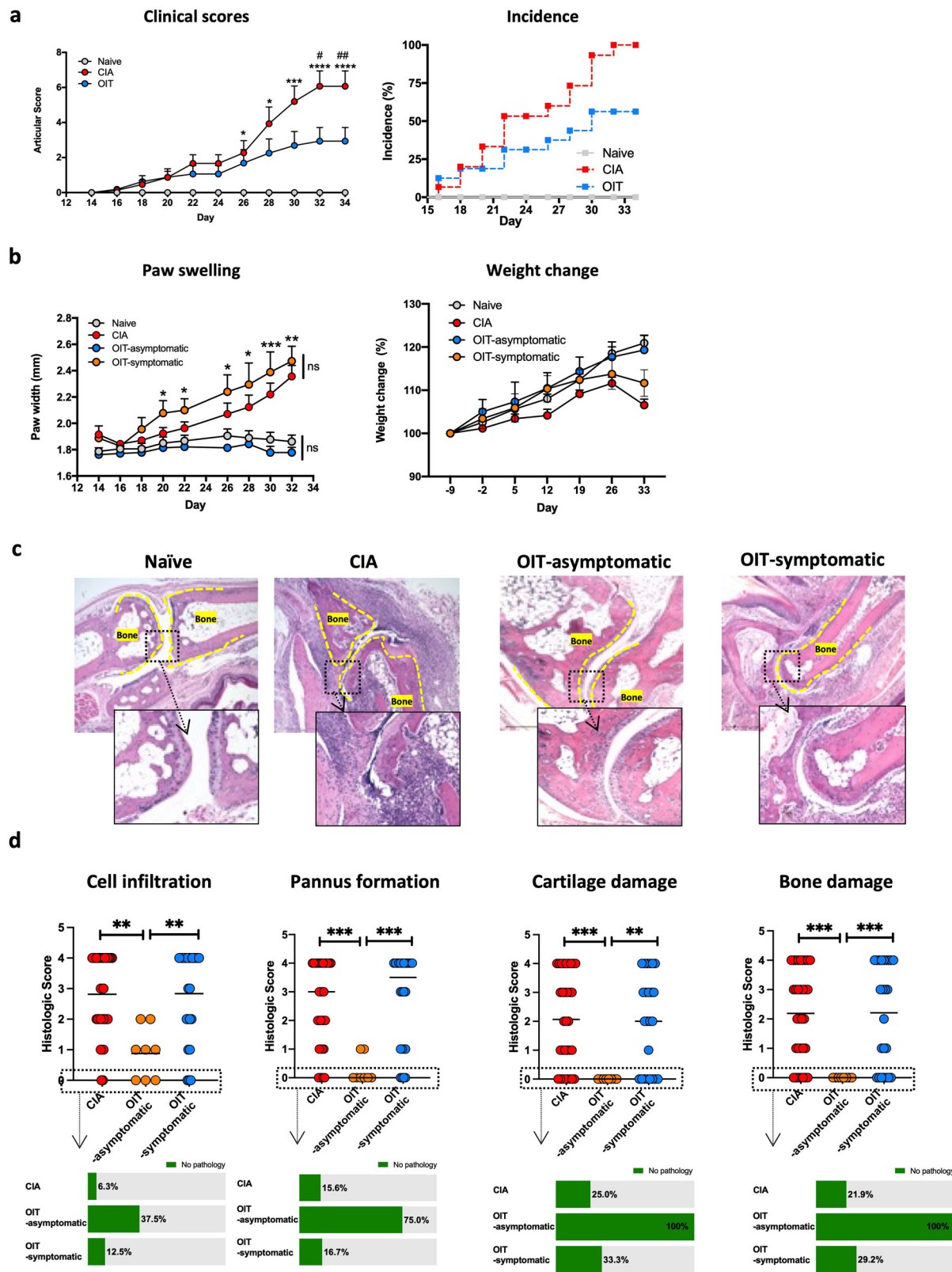

correlate with disease severity, regardless of treatment. The impact of B cell-derived IL-17 and IL-22 production on pathology was diminished compared to that of CD4$^+$ T cells, corroborating the leading role of Th17 cells in progressing joint inflammation. To corroborate these findings, we next analysed IL-17 and IL-22 expression in the joint tissue of CIA and asymptomatic OIT mice by immunofluorescence (Fig. 3c). In line with our previous results, non-inflamed tissue from naive mice exhibited minimal cytokine expression, whereas effective OIT mitigated the heightened expression observed in CIA, with a significant reduction in IL-22 (Fig. 3c).

OIT also reduced production of IL-17 by DLN cells from OIT mice, including both symptomatic and asymptomatic animals, compared with their CIA counterparts upon TPA/Ionomycin stimulation in vitro (Fig. 3d).

**Fig. 1 | Disease scores and histological analysis in response to oral immunotherapy (OIT) in CIA mice. a** Disease scores (left panel) and incidence (right panel) were evaluated at the indicated time points for naïve (grey), CIA (red) and CIA-OIT (blue). OIT consisted of administration of undenatured type II collagen by oral gavage 3 times a week, starting 2 weeks before induction of arthritis. Cumulative disease scores were given for each limb, 0 = no disease and 4 = highest score. Each dot represents the mean of individual mice and error bars show SEM ($n$ = 15 for CIA, $n$ = 16 for OIT groups, $n$ = 9 for naïve mice, data from 2 independent experiments). Statistical significance was determined by one-way ANOVA; *$p$ < 0.05, ***$p$ < 0.001 between CIA and naïve, #$p$ < 0.05, ##$p$ < 0.01 comparing CIA and OIT groups. Disease incidence was calculated by dividing the number of cases by the total number of mice in the group at the indicated times. **b** Paw width (left panel) and weight change (percentage of initial weight, right panel) were evaluated at the indicated time points. Each dot represents the mean of individual mice, error bars show SEM ($n$ = 15 for CIA, $n$ = 7 for OIT asymptomatic, $n$ = 9 for OIT symptomatic, $n$ = 9 for naïve mice,

data from 2 independent experiments). Statistical significance was determined by one-way ANOVA; *$p$ < 0.05, ***$p$ < 0.01,***$p$ < 0.001, ns = non-significant. **c** Hind paws were collected at the end of the experiment (day 34) for each mouse, sectioned, and stained with haematoxylin and eosin for histological analysis. Images show one representative mouse (Disease score = median for each group). Superimposed dotted lines show bone limits; scale bar = 500 µm. **d** Cell infiltration, pannus formation, cartilage and bone damage were quantified for individual mice in hematoxylin/eosin-stained sections. Dot plots show histological scores (from 0 to 4, 0 = healthy tissue, 4 = highest possible score). Individual dots represent individual paws ($n$ = 32 for CIA, $n$ = 8 for OIT asymptomatic, and $n$ = 24 for OIT symptomatic, data collected from 2 independent experiments). Bars show the mean value for each group. Statistical significance was determined by the Kruskal–Wallis test; **$p$ < 0.01, ***$p$ < 0.001. The percentage of mice with no sign of pathology (green) for the indicated parameter is shown in vertical column graphs.

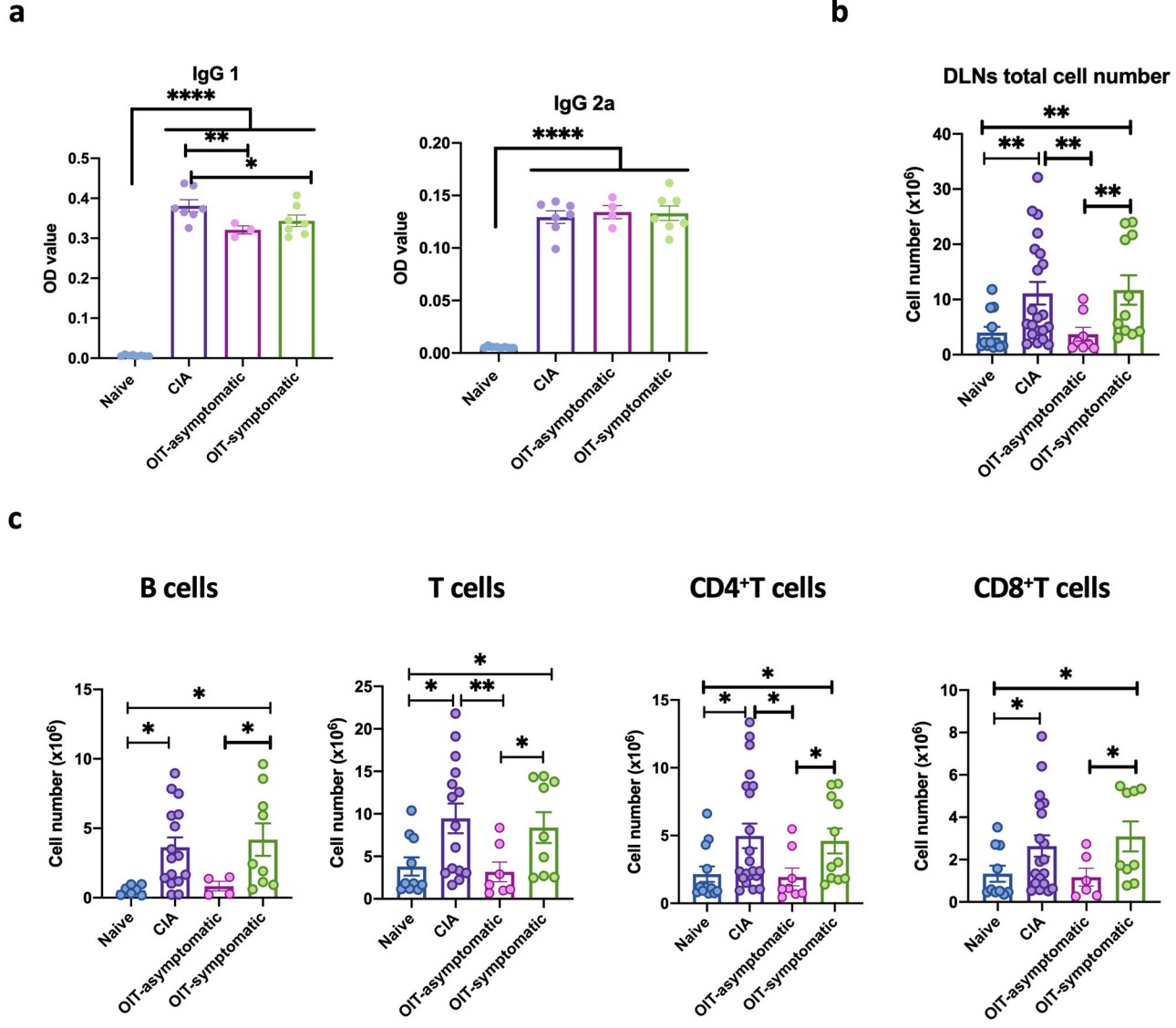

**Fig. 2 | Cellular and humoral responses in response to oral immunotherapy (OIT) in the joint. a** Anti type II collagen antibodies, IgG1 and IgG2 isotypes, were evaluated by ELISA in serum from naïve control mice ($n$ = 8), CIA ($n$ = 7, mean of disease score 4.86) and OIT mice ($n$ = 7 for OIT symptomatic, mean of disease scores = 7; $n$ = 4 for OIT asymptomatic) at day 33 after induction of arthritis. Values represent Optical Density (OD) at 450 nm. Each dot represents the mean values of individual mice analysed in technical triplicates. **b** Total cell numbers isolated from

draining lymph nodes (DLNs) collected from all naïve, CIA, asymptomatic and symptomatic OIT mice. **c** Total number of B cells, total T cells, CD4 and CD8 T cells in DLNs from (**b**) were evaluated by Flow Cytometry. Each dot in **a–c** column bars represents values of individual mice collated from 2 independent experiments. Error bars show mean ± SEM. Statistical significance was determined using ordinary one-way ANOVA. Significance is indicated by asterisks, *$p$ < 0.05, **$p$ < 0.01 and ***$p$ < 0.001.

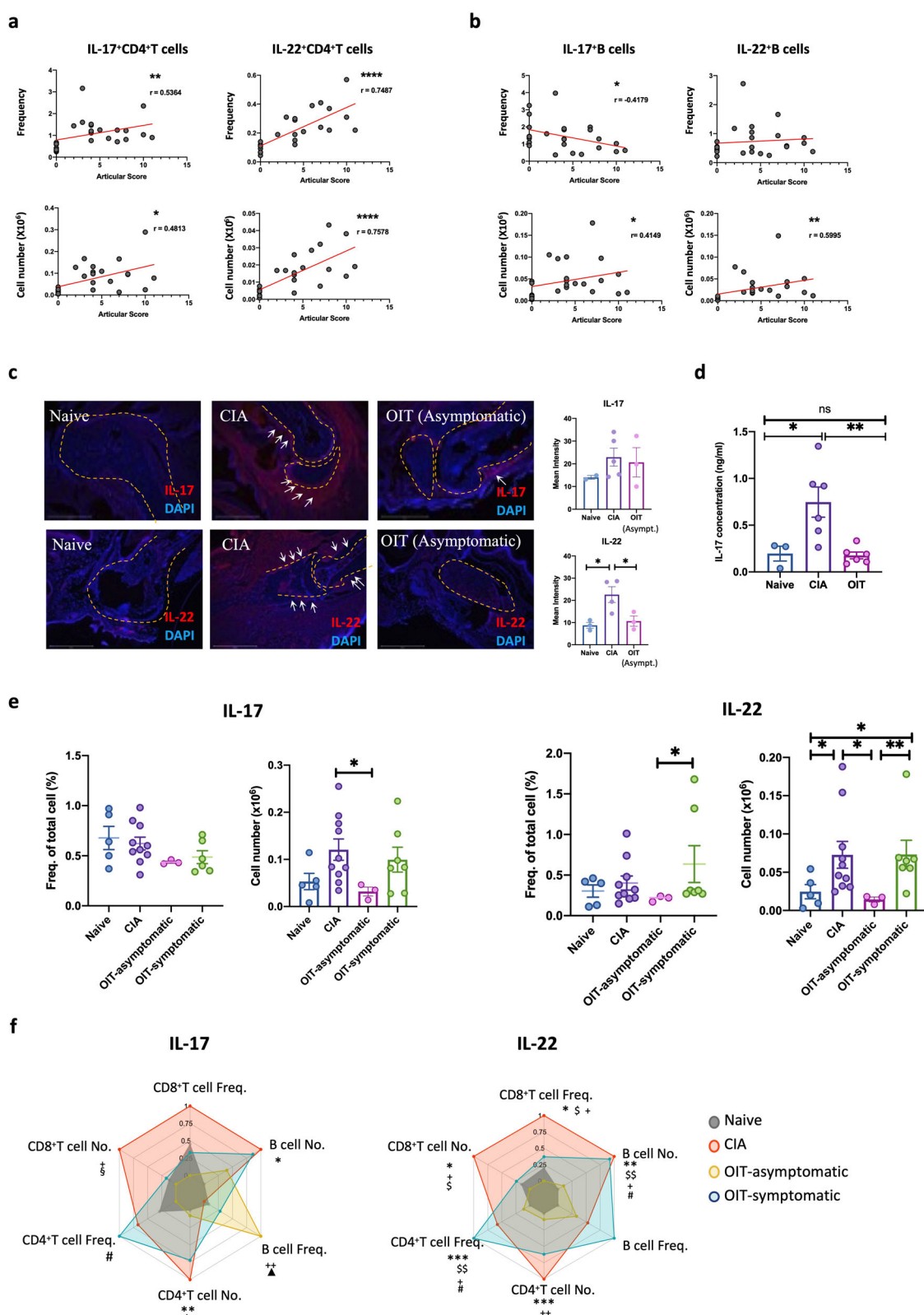

Following this, we conducted flow cytometry (Supplementary Fig. 3 for gating strategy) to investigate intracellular IL-17 and IL-22 expression in all DLN cells (Fig. 3e), categorising OIT mice into symptomatic and asymptomatic groups as before. Mice in the OIT group that remained asymptomatic exhibited markedly reduced expression levels of both IL-17 and IL-22 producers. Symptomatic mice displayed immune profiles that resembled those of arthritic mice, although they still showed a reduction, albeit not significant, in IL-17 producers. We further analysed which IL-17- and IL-22-producing DLNs were modulated in response to OIT. CD4[+] and CD8[+] T cells, and CD19[+] B cells data (cell frequency and numbers) were collated and normalised in radar charts for visualisation (Fig. 3f, raw data shown in Supplementary Fig. 4). This revealed that whilst IL-17[+] and IL-22[+] CD4 and

**Fig. 3 | OIT protection is associated with reduced upregulation of inflammatory IL-17 and IL-22 in the joint and draining lymph nodes.** Naïve, CIA and CIA-OIT mice were culled at day 33 when tissue was collected for further analysis.
**a, b** Correlation between numbers of IL-17 and IL-22 positive lymph node cells and clinical scores in T cells (**a**) and B cells (**b**) isolated from draining lymph nodes (DLNs). Every dot represents values for individual mice from all groups. Data are presented as mean ± SEM, *r*: Pearson's coefficient. **c** Expression of IL-17 and IL-22 (red) was evaluated in the joint tissue by immunofluorescence in naïve, CIA and OIT asymptomatic mice. DAPI (Blue) was used to stain nuclei as counterstaining. Superimposed dotted lines show bone tissue and areas of cell infiltration are indicated by white arrows. Scale bars: 500 μm. Graphs show the quantification of the mean intensity of individual mice. **d** IL-17 concentration was evaluated by ELISA in the supernatants of draining lymph node cells upon PMA (50 ng/ml)/Ionomycin (500 ng/ml) stimulation for 12 h. Data show naïve, CIA and OIT (symptomatic and

asymptomatic). Each dot represents cells from one individual mouse. Error bars show mean ± SEM; *p < 0.05, **p < 0.01 analysed by one-way ANOVA from one experimental model. **e** Relative frequency and total cell number of IL-17+ and IL-22+ DLN cells were evaluated by flow cytometry. Data show mean ± SEM; each dot represents individual mice from two independent experimental models; *p < 0.05, analysed by one-way ANOVA. **f** Cell frequency and total cell numbers of IL-17+ and IL-22+ CD4 T cells, CD8 T cells and B cells in DLNs, represented at the corners of radar charts: Naïve (grey), CIA (red) and OIT asymptomatic (orange) and OIT symptomatic (blue); data were normalised to maximum expression in each group. Significance on the raw data among groups was evaluated by ordinary one-way ANOVA, where *p < 0.05, **p < 0.01 [CIA vs naïve]; $^{\S}$p < 0.05 [naïve vs OIT symptomatic]; $^{\blacktriangle}$p < 0.05 [naïve vs OIT asymptomatic]; $^{+}$p < 0.05 [CIA vs OIT asymptomatic]; $^{\S}$p < 0.05 [CIA vs OIT asymptomatic]; $^{\#}$p < 0.05 [OIT asymptomatic vs OIT symptomatic].

CD8 T cells were significantly reduced, B cell-dependent production of IL-17 and, to a lesser extent, IL-22 were less affected in asymptomatic OIT relative to CIA mice. Moreover, an intriguing response was observed in asymptomatic OIT mice, which exhibited a significantly higher frequency of IL-17+ B cells (Fig. 3f, Supplementary Fig. 4).

## Oral administration of undenatured type II collagen protects against damage to gut tissue in CIA

The establishment of oral tolerance by feeding of antigens is a direct consequence of specific immune responses triggered in the gastrointestinal tract and gut-associated lymphoid tissue (GALT), which can lead to loss of mucosal barrier function in RA, suggesting that gut tissue architecture can modulate gut immunity and host-microbiome interactions[25]. Our previous work showed that gut pathology precedes and perpetuates chronic systemic inflammation driving autoimmunity and joint damage in CIA[24]. Therefore, we evaluated the integrity of the gastrointestinal tract (duodenum, jejunum, ileum and colon) in our experimental OIT model (Fig. 4a). CIA mice showed substantial damage in all gut areas, such as disruption of the epithelial layer, cell infiltration and thickening of the muscular layer (Fig. 4b). We quantified the ratio of villi to crypt length in the gut to compare the health and function of the intestinal mucosa. CIA mice exhibited a diminished villi-to-crypt ratio, reminiscent of findings in inflammatory gut conditions. This alteration was rectified in the duodenum of asymptomatic OIT mice, with a similar trend in the jejunum and ileum. This effect was not observed in the colon. Interestingly, symptomatic OIT mice, characterised by joint pathology, also exhibited an absence of gut damage or pathology. Indeed, symptomatic OIT tended to exhibit even higher villi-to-crypt ratios than healthy mice, particularly in the distal areas of the small intestine such as the ileum but not in the colon (Fig. 4b). Given the persistently reduced villi-to-crypt ratio observed in the colon of all OIT mice, we conducted Periodic Acid-Schiff (PAS) staining to visualise further tissue damage associated to redistribution or changes in glycogens and mucins. PAS staining revealed attachment/effacement lesions in the colon of arthritic mice (Fig. 4c), in line with previous reports[24,25]. However, such lesions were absent in asymptomatic OIT mice but not in those showing joint symptoms (Fig. 4c).

Finally, to assess how the safeguarding of gut tissue in asymptomatic OIT mice translated into gut immune responses, we examined the total number of cells in mesenteric lymph nodes (MLNs), as well as those of B cells and CD4+ and CD8+ T cells. We did not observe any significant difference in the number of DLN cells amongst these experimental groups. Nonetheless, both asymptomatic and symptomatic OIT mice exhibited an expansion of B cells in mesenteric lymph nodes (MLNs), not only in comparison to the CIA group but also in relation to the naive controls, albeit statistical significance was attained only for the symptomatic group (Fig. 4d). In contrast, the numbers of CD4+ and CD8+ T cells in MLNs remained unaltered in both cases.

Overall, our results indicate that OIT modulates gut immunity and triggers protective mechanisms to preserve gut integrity, even in those mice who ultimately develop joint pathology. Hence, our next objective was to

conduct a comprehensive characterisation of the local immunological pathways responsible for protection. We, therefore, worked with asymptomatic OIT mice, a cohort demonstrating neither joint nor gut pathology. We used cells from MLNs and cells isolated directly from the gut, following enzymatic tissue digestion (Gating strategy shown in Supplementary Fig. 5). In line with our experiments in the joint tissue (Fig. 3), we investigated IL-17 expression by flow cytometry in the mesenteric lymph nodes (MLNs) and ileum and colon tissue (Fig. 5), to assess correlations between gut cellular networks with distant responses within the joint. Contrary to the DLN data (Fig. 3), the levels of IL-17 producers in total MLN cells (Fig. 5a), ileum (Fig. 5b) or colon (Fig. 5c) tissue were not significantly elevated in CIA mice. Perhaps surprisingly, there was a trend towards reduced levels of IL-17+ cells in the ileum, although there tended to be higher levels of these cells in the colon of CIA mice, inverse patterns that were reversed in asymptomatic OIT mice (Fig. 5b, c). However, analysis of specific cell types revealed a distinct rewiring of the IL-17-producing networks associated with healthy animals, in both the CIA and asymptomatic OIT groups, in each of the MLNs (Fig. 5d), ileum (Fig. 5e) and colon (Fig. 5f) tissue. Overall, a broad analysis indicates that in healthy conditions, IL-17 is generally produced by innate cells (NKT, ILC3 and γδ T cells), whilst in CIA this predominantly switches to adaptive CD4/CD8 cell responses in MLNs, and expansion of more specialised cell types in the gut tissue, with OIT showing a mixed pattern distinct to both naive and CIA networks (Raw data from radar charts are shown in Supplementary Fig. 6). Intriguingly, OIT protection tended to correlate with increased levels of IL-17+ producers in the ileum, although the results did not reach statistical significance. Moreover, their production of IL-17 showed a general increase in the OIT group when assessing the mean fluorescence intensity in all cell types.

We conducted a similar approach to evaluate IL-22 expression in MLNs and gut tissue (Fig. 6), bearing in mind that IL-22 exerts key mucosal healing mechanisms, promoting epithelial regeneration and integrity, mucus production and synthesis of antimicrobial peptides[34–36]. We did not observe any significant differences among groups in total IL-22+ MLNs (Fig. 6a), or in the cells isolated from the ileum (Fig. 6b) or colon (Fig. 6c). Nevertheless, and in line with the previous results shown for IL-17, flow cytometric analysis of the distinct populations of producers suggests that IL-22 networks are actively rewired in asymptomatic OIT mice compared to CIA and naïve controls. Thus, production of IL-22 in naïve MLNs comes mostly from B cells and innate ILC3 and NKT cells whilst in CIA mice it extends to CD4+ T cells, NKTs and γδ T cells, the latter being the only enhanced populations in asymptomatic OIT (Fig. 6d). Analysis of IL-22 mean fluorescence intensity revealed a generalised increase in IL-22 production in MLNs during CIA, whereas this was reduced both in the ileum (Fig. 6e) and colon (Fig. 6f), thereby corroborating the protective role of local IL-22 in the gut tissue. In fact, and in contrast to the IL-22 results in MLNs, CIA mice did not increase IL-22 production, whilst all innate populations located in the gastrointestinal tract increased the expression of IL-22 in asymptomatic OIT mice, as measured by mean fluorescence intensity (MFI) (Fig. 6e, f) (Raw data from radar charts are shown in Supplementary Fig. 7). Although we cannot unequivocally define intestinal

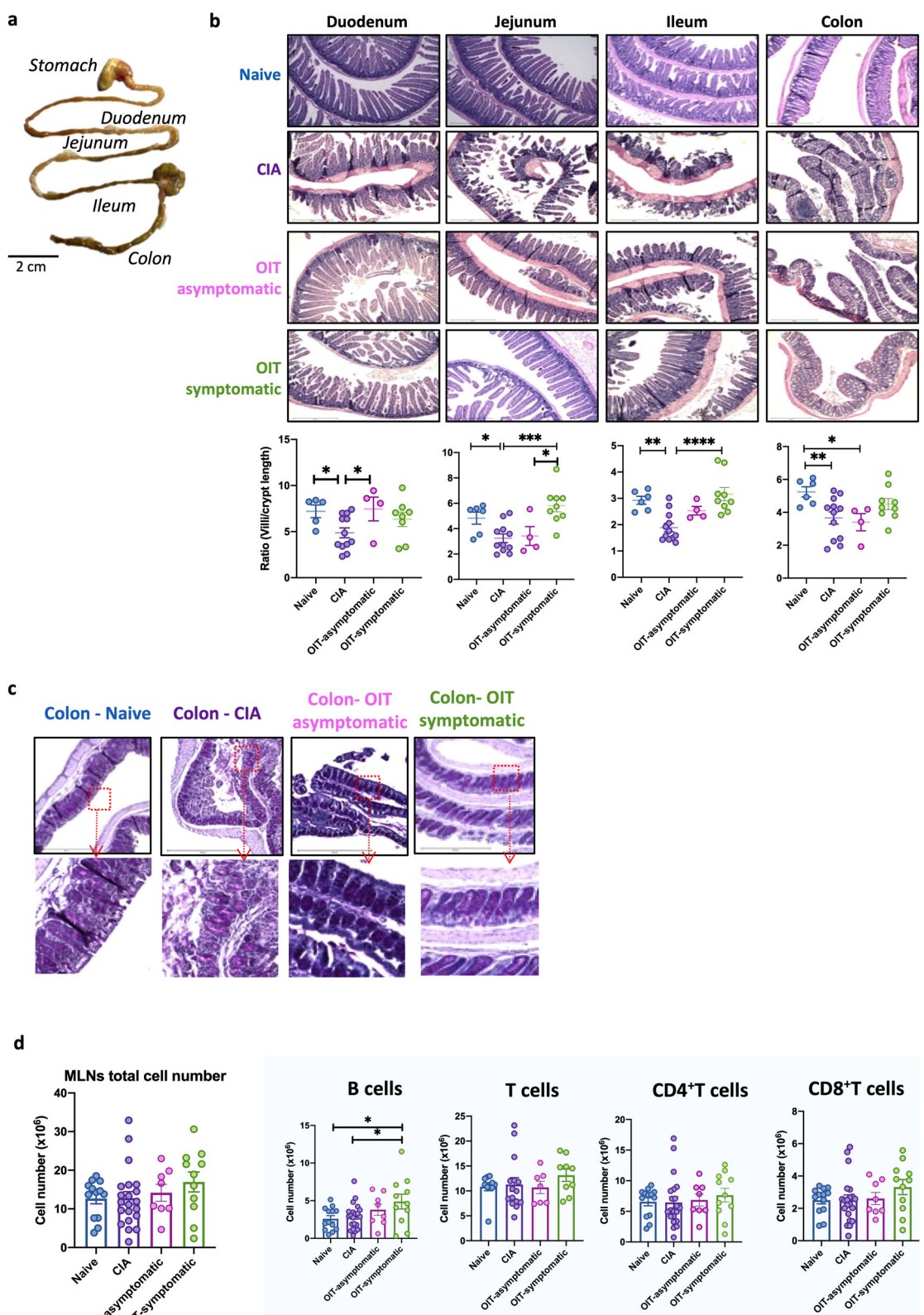

IL-22 as a protective or pathogenic factor, our results highlight that differential rewiring of the IL-22$^+$ cell network in the gut is associated with inflammatory or homoeostatic conditions, with a general expansion of IL-22$^+$ innate populations and significant increase in IL-22 production by NKT cells in asymptomatic OIT mice. Finally, IL-22 staining in gut tissue provided further support for the protective role of local gut IL-22 expression in OIT mice, as there was a significant increase in IL-22 secretion/deposition in the tissue epithelium in the colon of asymptomatic OIT mice compared to that CIA mice (Fig. 6g, h). Interestingly, increased staining of IL-22 was also observed in symptomatic OIT mice (Fig. 6g, h), perhaps explaining why the gut of symptomatic OIT mice also exhibited protected tissue integrity (Fig. 4b, c).

**Fig. 4 | OIT protects against gastrointestinal damage associated with inflammatory arthritis in both symptomatic and asymptomatic cases.** Naïve, CIA and OIT asymptomatic mice and OIT symptomatic mice were culled at day 33 when gut tissue and total mesenteric lymph nodes (MLNs) were collected. **a** Isolated gastrointestinal tract from a naïve mouse showing the four anatomical areas used for further study. **b** Duodenum, jejunum, ileum, and colon were fixed, and tissue sections were subjected to hematoxylin and eosin staining. The length of villi and crypts was measured using Image J software, and the ratio of villi/crypt was quantified. Each dot represents the villi/crypt ratio for individual mice, and data were collated from two independent experiments. Statistical significance was evaluated by ordinary one-way ANOVA, where $*p < 0.05$, $**p < 0.01$ and $***p < 0.001$. **c** Colon tissue sections were stained with PAS to detect changes in the mucus layer and associated pathology. Scale bars = 500 μm. The depicted mice are representative of individual mice whose disease scores fall within the median value for each group. **d** MLNs were collected to generate single-cell suspensions, and a total number of cells was counted. Cell number and percentage of B cells, total T cells, CD4 and CD8 T cells were evaluated by Flow Cytometry. Each dot represents one individual mouse and error bars show mean ± SEM. Mice were pooled from 2 independent experiments. Statistical significance was determined using Ordinary one-way ANOVA. Significance is indicated by asterisks, $*p < 0.05$, $**p < 0.01$.

## Gut fucosylation and microbiome composition are reshaped in OIT and arthritic mice

Since our results suggest that intestinal IL-22 networks correlate with effective OIT, we next investigated potential protective mechanisms triggered by this cytokine in asymptomatic OIT mice. In health, IL-22 not only supports homeostatic expression of mucins in epithelial cells[37], but also their fucosylation stage[38]. Fucosylation is a specific type of post-translational glycosylation, that prevents infections and enhances the integrity of the epithelial gut layer[39,40]. Therefore, because asymptomatic OIT mice preserved gut tissue integrity and had enhanced gut IL-22 expression, we first hypothesised that such protection was associated with increased epithelial fucosylation, which would, in turn, prevent gut damage during inflammation. To investigate this, we stained the tissue with two lectins that recognise fucosylated glycans, *Ulex european* Agglutinin (UEA) and *Auleria aureate* lectin (AAL), that bind to terminal and core fucosylation respectively. Consistent with the protective role proposed for terminal fucosylation, UEA binding (recognising terminal alpha[1,2] linked fucose residues) was significantly reduced in the ileum of CIA mice, while asymptomatic OIT mice more resembled the profile of healthy tissue, as they were not significantly different to the Naïve group (Fig. 7a). By contrast, no significant difference was seen in the colon across the groups in this model (Fig. 7a). Moreover, neither the ileum nor the colon, showed significant differences in core fucosylation as detected by AAL binding (Fig. 7b). Next, we evaluated mRNA expression of fucosyltransferases (FUTs) in whole gut tissue, enzymes responsible for fucosylated glycan biosynthesis. In the ileum, only expression of FUT8 mRNA was significantly different in asymptomatic OIT mice (Fig. 7c), whereas no significant changes were seen in the colon (Fig. 7d). These results in FUT expression do not explain the observed maintenance of fucosylation in OIT mice compared to the reduced levels in CIA mice (Fig. 7a), suggesting that other factors may be involved. This is not completely unexpected since gut epithelial fucosylation is strongly dependent on environmental factors, such as microbiome composition, which is also regulated by cytokine-dependent mechanisms[41]. Dysregulation of some microorganisms in the gut has been linked to RA pathogenesis and morbidity in multiple studies, perhaps through subsequent variation of intestinal metabolites that promote inflammation in the target tissue[42,43]. Therefore, an alternative hypothesis is that the rewiring of IL-17/IL22 immune networks upon undenatured type II collagen administration modifies the composition of the gut microbiome, or vice versa, which in turn, can affect mucin fucosylation consolidating dysregulation of the microbiome to perpetuate systemic inflammatory response. To provide support for this hypothesis, we conducted 16 S amplicon sequencing to investigate the microbial diversity in the ileum (Fig. 8) and colon (Fig. 9), using faecal samples from naïve, CIA and OIT mice. As before, OIT mice were separated into symptomatic (disease scores 5 ± 0.6, $n = 3$) and asymptomatic mice. We also separated CIA mice into established disease with high scores (9.3 ± 2.8, $n = 3$), and mice with more recently initiated joint inflammation and lower disease scores (3.6 ± 0.22, $n = 3$), to identify changes in microbial content led by OIT and not a reduced inflammatory environment. To understand the microbial community diversity in the samples (Within-community) we looked at α-diversity indexes Dominance, Simpson, Observed_otus, Shannon and Chao1 (Fig. 8a). The ileum microbiome of OIT asymptomatic group showed significant differences in the Dominance (higher) and Simpson index (lower), suggesting a microbial community that is characterised by lower diversity, skewed abundance distribution, and less evenness across different taxa. Differences in β-diversity visualised by the UniFrac distance suggest that the microbial communities in the OIT asymptomatic mice have distinct compositions (Fig. 8b). The composition and relative abundance of the ileum microbiota at the phylum level were examined (Fig. 8c). Firmicutes was the dominant phylum in all groups, followed by Bacteroidetes (Bacteroidota). CIA mice at earlier joint disease stages (CIA low) exhibited a fivefold reduction in the proportion of the Bacteroidota, a loss that was absent in both symptomatic and asymptomatic OIT mice. Perhaps unexpectedly, CIA mice exhibiting more established, high-score arthritis, rather than the more recently developed low-score arthritic mice, displayed microbiota profiles and indexes closer to those of naïve mice. This is likely related to the appearance of self-resolving mechanisms described in the CIA model that can occur around two weeks after joint disease onset such as TGF production[44], which can affect the microbiome composition, particularly the Bacteroidetes[45], thereby suggesting that CIA mice with lower scores due to them being at earlier stages of disease may provide a better reference for identification of pathogenic microbiome changes. Compared to symptomatic mice, the asymptomatic OIT group increased the diversity of bacteria in the colon, expanding the Campylobacteria and Proteobacteria phyla. Moreover, the analysis of the top 35 genera in abundance (Fig. 8d) revealed that whilst the *Lachnospiraceae*, *Mucispirillium*, and *Anaerotruncus* showed a consistent reduction in the ileum of asymptomatic OIT mice, *Muribaculaceae* species were increased (Fig. 8e).

A similar analysis was carried out on the colon faecal material, specifically showing alpha diversity indexes (Fig. 9a), beta diversity by UniFrac distance (Fig. 9b), and relative abundances at the phyla (Fig. 9c) and genera (Fig. 9d, e) levels. Contrary to the ileum, the colon microbiome of OIT asymptomatic mice presents a lower dominance index and a higher Simpson index compared to symptomatic OIT and arthritic mice with lower disease scores (Fig. 9a), suggesting a more diverse and even population that is not strongly dominated by a few specific taxa in asymptomatic OIT. Consistent with previous reports, analysis at the phyla level reveals higher diversity in the colon than in the ileum in naïve mice. Firmicutes remain the most abundant phylum, but there is an increased proportion of Bacteroidetes and Deferribacteriota. OIT asymptomatic mice showed a more diverse profile overall but exhibited a reduction in the Bacteroidetes and Patesciabacteria phyla relative to the CIA low-score arthritic mice (Fig. 9c). At the genus level, the abundance of the top 35 genera present changed between asymptomatic OIT mice and the rest of the groups (Fig. 9d). *Clostridia_vadinBB60* were reduced, whereas others like *Colidextribacter*, *Roseburia*, *Rikenella*, and *Ruminococcaceae* were increased, relative to the CIA-low score group (Fig. 9e). Critically, the distinct profiles of healthy (Naïve) and OIT asymptomatic mice suggest that rather than simply preventing CIA-induced changes in the microbial species, OIT rewires the interactive network between the gut immunoregulatory pathways and commensal bacteria to a phenotype that protects against arthritis. Such a network is likely to be bidirectional, with bacterial metabolites further adjusting host immunity. To gain some insight into the potential biological activities of microbial communities without directly measuring gene expression or protein function, we conducted some metagenomic function prediction based on the 16 S sequencing using the bioinformatic tool PICRUSt2[46]. This analysis indicated differences among naïve, CIA and OIT groups, particularly in the ileum of OIT asymptomatic mice

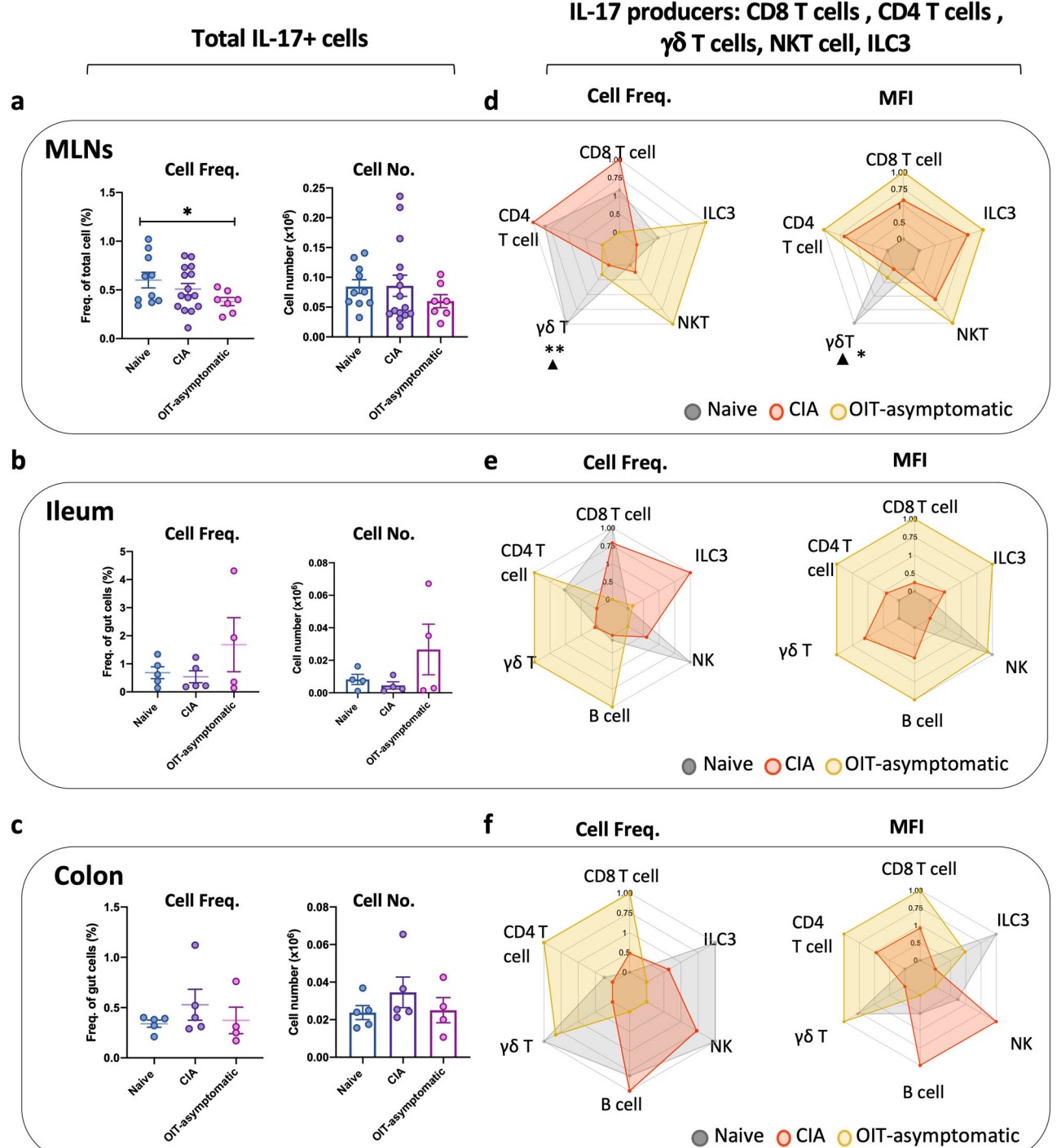

**Fig. 5 | Expression of IL-17 in mesenteric lymph nodes gastrointestinal tract.** Naïve, CIA and OIT asymptomatic mice were culled at day 33, when mesenteric lymph nodes (MLNs), ileum and colon samples were collected. Single-cell suspensions were obtained from MLNs and digested gut tissue, and IL-17 expression was subsequently evaluated by flow cytometry in total isolated cells (**a–c**) and specific cell populations, including CD4 T cells, CD8 T cells, group 3 innate lymphoid cells (ILC3), γδ T cells and NK cells (**d–f**). **a–c** Percentage and number of total IL-17+ cells in MLNs (**a**), ileum samples (**b**) and colon (**c**). Each dot represents values of individual mice; bars show mean values for each group ± SEM; Naïve *n* = 10. CIA

*n* = 15, asymptomatic OIT *n* = 7. Statistical significance was determined using ordinary one-way ANOVA, *$p < 0.05$. **d–f** Relative cell frequency and mean fluorescence intensity (MFI) of IL-17 in the indicated cell populations in MLNs (**d**), single cells isolated from the ileum (**e**) and colon (**f**). Each corner of the radar charts represents the indicated normalised parameter for naïve (grey), CIA (red) and asymptomatic OIT (orange) mice. Data were normalised to maximum expression in each group; naïve *n* = 5, CIA *n* = 5, asymptomatic OIT *n* = 4. Statistical significance was determined using raw data and ordinary one-way ANOVA. *$p < 0.05$, **$p < 0.01$ in CIA versus Naïve; ▲$p < 0.05$ [naïve vs OIT asymptomatic].

(Supplementary Fig. 8a), but also in the colon (Supplementary Fig. 8b). Although further experimental work is required to demonstrate this, this data suggest that effective OIT could impact of the functional host-microbiome crosstalk during chronic arthritis.

## Discussion

Oral tolerance mechanisms have been the subject of investigation since the beginning of the 20th century, but translation of these findings into a clinical setting has not always been successful, indicating that key pathways are still

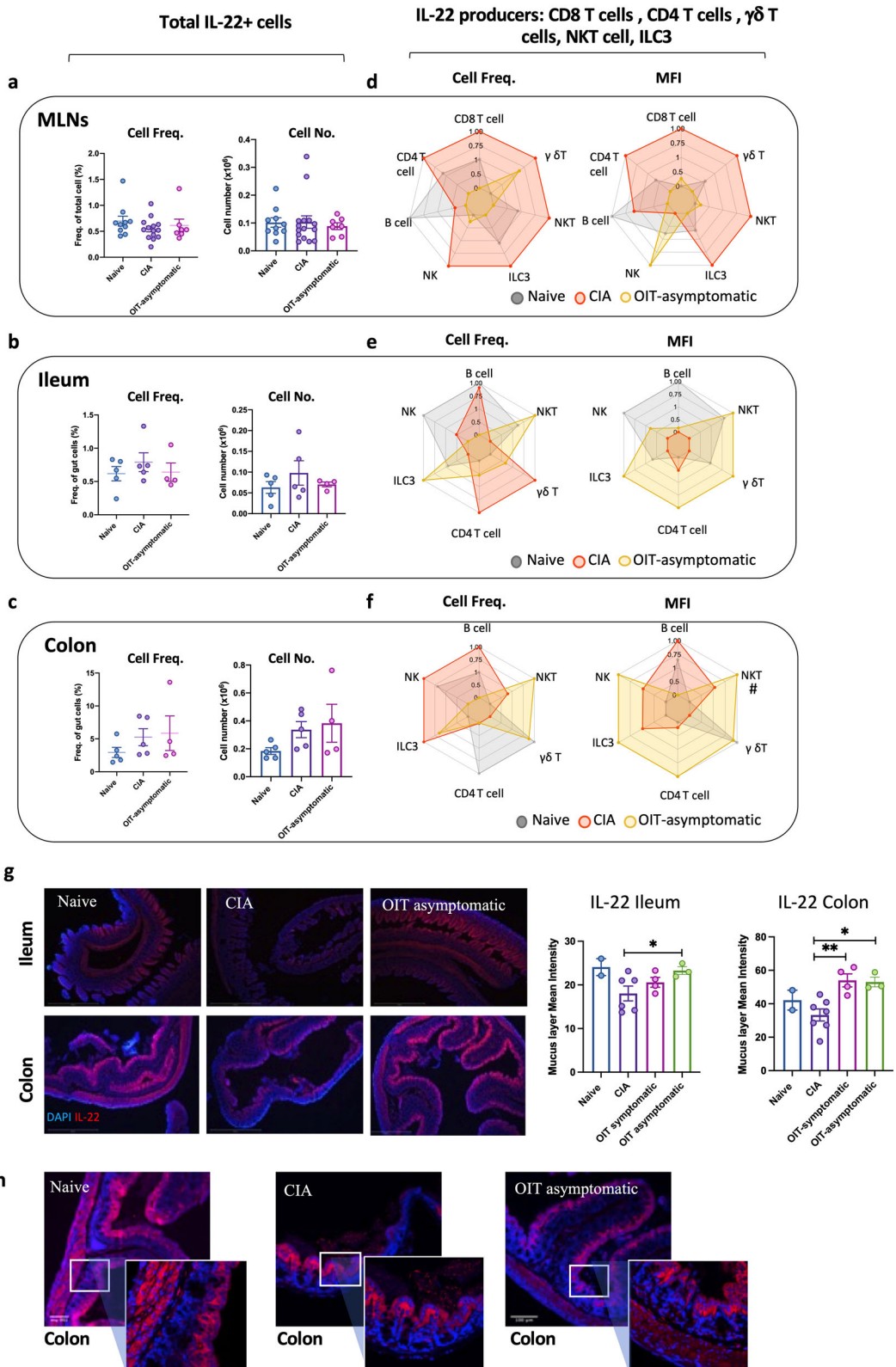

unknown. Our results show that a high percentage of animals (~55%) that received OIT were fully protected against experimental arthritis, with no symptoms of disease, supporting the previously described protective role of undenatured type II collagen in inflammatory joint disease[15,22,47–49]. Our analysis of asymptomatic and symptomatic OIT mice, in comparison with the CIA group, therefore, contributes to our understanding of the

mechanisms triggered by OIT to achieve a therapeutic effect. Asymptomatic OIT mice are strongly protected against cartilage and bone damage, which may be a direct consequence of the lower IL-17 and IL-22 levels in the joint. We still observed some cell infiltration in these mice, however, suggesting that this was not sufficient to initiate any inflammatory response or reflected recruitment of inflammation resolving/suppressing cells. By contrast,

**Fig. 6 | Expression of IL-22 in mesenteric lymph nodes and gastrointestinal tract.** Naïve, CIA and OIT asymptomatic mice were culled at day 33, when mesenteric lymph nodes (MLNs), ileum and colon samples were collected. Single-cell suspensions were obtained from MLNs and digested gut tissue, and IL-22 expression was subsequently evaluated by flow cytometry in total isolated cells (**a–c**) and specific cell populations, including CD4 T cells, CD8 T cells, B cells, group 3 innate lymphoid cells (ILC3), γδ T cells, NK cells and NKT cells (**d–f**). **a–c** Percentage and number of total IL-22+ cells in MLNs (**a**), ileum samples (**b**) and colon (**c**). Naïve n = 10. CIA n = 15, asymptomatic OIT n = 7. Each dot represents values for individual mice; bars show mean values for each group ± SEM. **d–f** Relative cell frequency and mean fluorescence intensity (MFI) of IL-22 in the indicated cell populations in MLNs (**d**),

single cells isolated from the ileum (**e**) and colon (**f**). Each corner of the radar charts represents the indicated normalised parameter for naïve (grey), CIA (red) and asymptomatic OIT (orange) mice. Data were normalised to maximum expression in each group; naïve n = 5, CIA n = 5, asymptomatic OIT n = 4. **g** Ileum and colon sections were stained with anti-IL-22 antibodies and specific secondary antibodies (Red) and DAPI (Blue) as counterstaining. Scale bars = 500 μm. Pixel intensity for IL-22 staining was quantified using ImageJ, each dot shows the mean value of 10 different areas for each individual mouse. Error bars show standard error (SEM). Statistical significance was determined using raw data and ordinary one-way ANOVA, where *$p < 0.05$, **$p < 0.01$. **h** Detailed images of colon sections stained for IL-22, scale bars = 100 μm and 20 μm for zoomed areas.

symptomatic OIT animals exhibited immune cell infiltration and clinical joint scores similar to those of CIA mice, suggesting that OIT impacted disease incidence rather than disease severity. Nevertheless, both symptomatic and asymptomatic mice showed protection against gut damage, indicating that all OIT mice had undergone some local immunoregulation regardless of whether they developed clinical joint symptoms or not. Such gut local protection could be a consequence of the elevated IL-22 gut levels observed in both groups. IL-22 promotes the protection of the gut tissue, primarily through the maintenance of mucosal homoeostasis and activation of anti-microbial responses[50]. Thus collectively, the high IL-22 in the joint and gut of symptomatic OIT mice may reflect pro-inflammatory effects in the joint, but homeostatic mechanisms in the gut. IL-22, therefore, could play a central role in OIT, a factor that has been involved not only in joint inflammation and epithelial repair, as discussed, but also in the regulation of fucosylation of mucins and modulation of the gut microbiome, factors that are also modulated in our experimental model. However, we still do not know the factors that ultimately drove protection in the asymptomatic OIT mice. Environmental factors could be affecting the functional outcomes of IL-22 in some mice, and further studies should address these to harness all the therapeutic potential of undenatured type II collagen.

Following the comparative analysis of symptomatic and asymptomatic OIT mice, we next focused on the investigation of the protective mechanisms activated in mice with no clinical symptoms. This demonstrated that the protective effect of undenatured type II collagen occurs, at least in part, through the protection of targeted organs distal to the joint such as the gut, where it acts broadly to rewire cytokine networks and regulate the gut dysbiosis observed in arthritic mice. Interestingly, asymptomatic OIT mice displayed a distinct ileum and colon microbiome rather than a preservation of the bacteria found in healthy animals, indicating an active modulation of the bacterial community and potentially, the subsequent metabolite production to effect protection. Interestingly, therefore, *Roseburia* species were expanded in asymptomatic OIT, a genus which produces immunoregulatory Short Chain Fatty Acids (SFCA) and inhibits Th17 but increases IL-22 responses in the gut[51–54]. These actions overlap with our in vivo findings in the intestine of asymptomatic OIT mice, providing support to the hypothesis that rewiring of functional microbiome-cytokine networks is boosted in protected OIT. Such networks will likely include more players, increasing the level of complexity and bidirectional regulation. For example, Ruminococcaceae, another genus increased in the colon of asymptomatic OIT mice, produces SFCA which has been directly implicated in activating the Wnt/β-catenin pathway responsible for gut epithelium regeneration[55]. Likewise, modulation of local gut immune system populations in OIT may exert some selective pressure on the microbiome composition, to facilitate the dominance of protective genera. For example, the expansion of ILC3 cells observed in asymptomatic OIT mice could contribute to this by modulating the microbiota composition via epithelial fucosylation[40,41]. Additional experiments are needed to provide further support to these data, but our findings provide additional evidence for the role of a pathogenic gut–joint axis in arthritis and describe new networks that may underpin the induction of anti-inflammatory responses upon exposure to fed antigens such as undenatured type II collagen.

Amelioration of joint inflammation and bone damage during protective OIT was also associated with the downregulation of pro-inflammatory

IL-17 in the joint and draining lymph nodes. Inflammatory factors produced systemically and in the joint, such as IL-17, could induce damage in the gut tissue to exacerbate disease pathology, but a breach of tolerance at mucosal surfaces is thought to be an initial event in RA that can occur many years before disease onset. Which factor is the cause or effect is not known, although both scenarios appear to happen in specific clinical cases. Moreover, we and others have shown that microbiome dysbiosis and gut pathology precede systemic inflammation, autoimmunity and joint disease in the CIA model[24,56]. In any case, gut dysbiosis and metabolite production seem to be a driving factor to ultimately lead to disease progression in mouse models and human studies[57,58].

Some well-defined tolerogenic mechanisms did not appear to be critical to protection in OIT mice, including the reduced production of anti-collagen antibodies or induction of regulatory T cells (Treg) in the gut and draining lymph nodes, although it has previously been reported that type II collagen increased levels of Treg and TGFβ, in addition to the reduction of IL-17[59]. This lack of effect on Tregs can be a consequence of the antigen dose used in our study since Treg activity is stimulated by lower doses[60]. The dose of 7.33 mg/kg was chosen because our preliminary studies suggested this to be more efficient under our experimental conditions, perhaps because of the strong inhibitory effect of IL-17-dependent pathways which are the main drivers of CIA. A dose of 2 mg/kg was efficient in a recent study of experimental osteoarthritis, both in young and older animals[49]. Our study focused on the late stages of the disease, and we cannot rule out that Tregs are involved prior to the induction of IL-17-mediated pathways. Likewise, we did not evaluate the numbers of regulatory B cells (Bregs), but there is experimental evidence showing that aberrant regulation of this cell type modulates arthritis progression via gut-dependent mechanisms[24,25]. In fact, we observed a significant increase in the number of B cells in the gut in response to undenatured collagen treatment but no effects in the production of anti-collagen antibodies, perhaps indicating a role for Bregs, an area we plan to address in follow-up studies. For example, since Tregs are generated in the gut during the induction phase of disease[61], additional studies, including pre-clinical disease stages are required to provide a full understanding of undenatured type II collagen-dependent regulatory responses. The impact of oral administration of undenatured type II collagen on B cells in MLNs prior to the initiation of disease might also be needed to fully understand its protective mechanisms, although due to the lack of lineage-specific markers, these experiments will be more challenging to conduct.

Reduction of IL-17 can be directly related to bone protection, as it increases RANKL and osteoclastogenesis[62]. Interestingly, IL-22 expression in draining lymph nodes and joints was increased in CIA mice and decreased in the asymptomatic OIT group (both IL-17+/IL-22+ CD4 T cell levels show positive correlations with increasing articular score). This is contrary to the observations in the gut, where IL-22 was generally up-regulated in OIT mice and provides further support for the dual role described for IL-22 during arthritis[31,63], perhaps related to the ability of this cytokine to promote gut epithelial cell regeneration[50]. Thus, IL-22 can be a critical contributor to both disease pathology and protective responses, as the newly defined gut–joint axis theory[64] proposes that the gut mucosa is the first site of disease initiation. Consistent with this, CIA mice displayed significant gut pathology and structural damage that was absent in OIT mice. IL-17 acts to regulate mucosal host defence against many invading

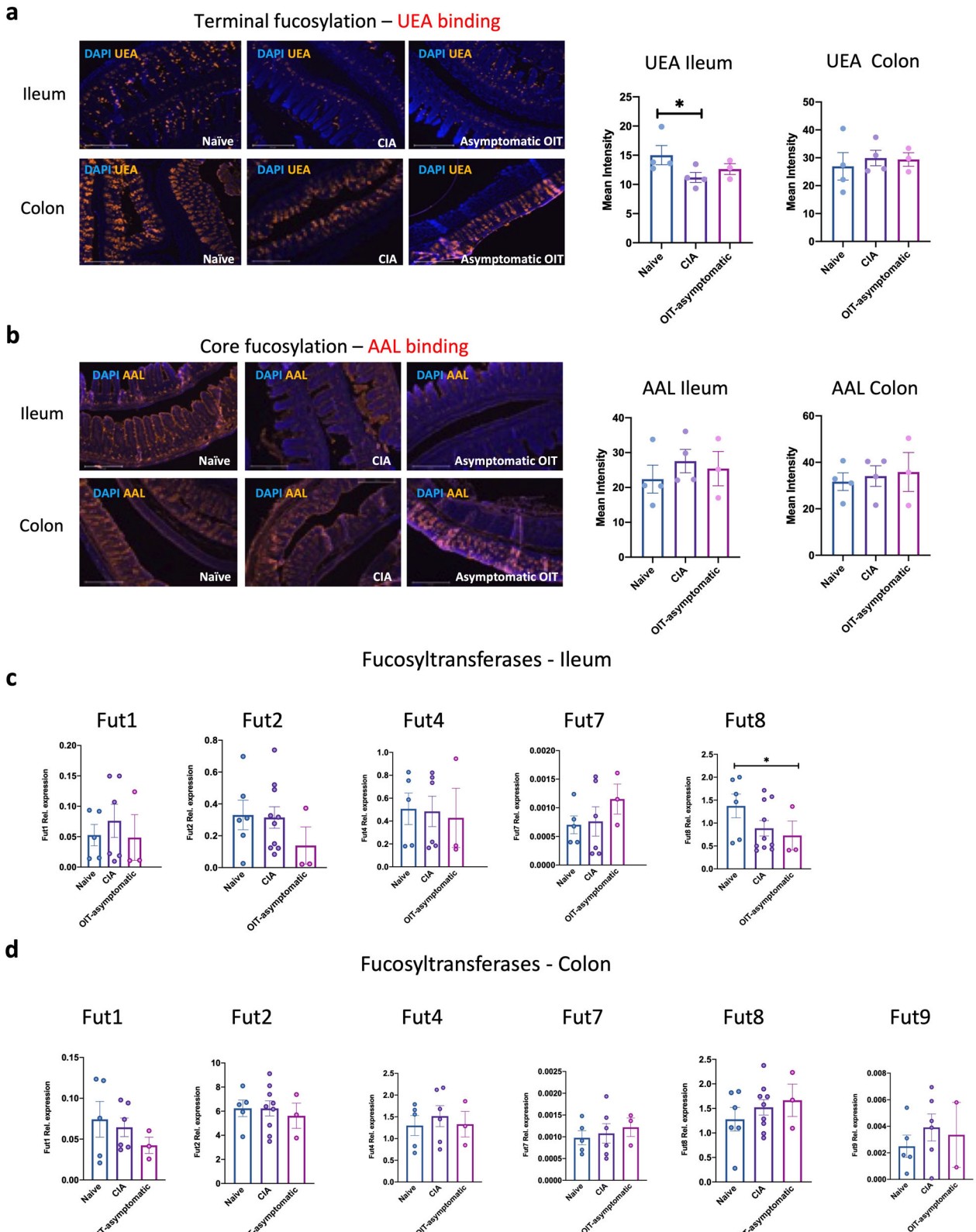

**Fig. 7 | Effect of protective OIT on fucosylation and fucosyltransferase expression in the gut tissue during arthritis. a, b** Ileum and colon sections from naïve, CIA and asymptomatic OIT mice were stained with UEA (**a**) and AAL (**b**) biotinylated lectins and fluorescence streptavidin (yellow) to detect terminal and core fucosylation respectively. DAPI (Blue) was used as counterstaining. Scale bars: 500 μm. Images show one representative example of each group. Graphs show the mean pixel intensity for lectin staining in individual mice; each dot shows the mean value from 10 different analysed areas, quantified using Image J software. **c, d** Relative expression of fucosyltransferases mRNA was evaluated by RT-PCR in the ileum (**c**) and colon (**d**) samples, including FUT1, FUT2, FUT4, FUT7, FUT8 and FUT9. Expression is shown as relative to actin expression. Statistical significance was evaluated by one-way ANOVA where *$p < 0.05$. Each dot represents values of individual mice where data are collected from two independent experiments. Error bars represent standard error (SEM).

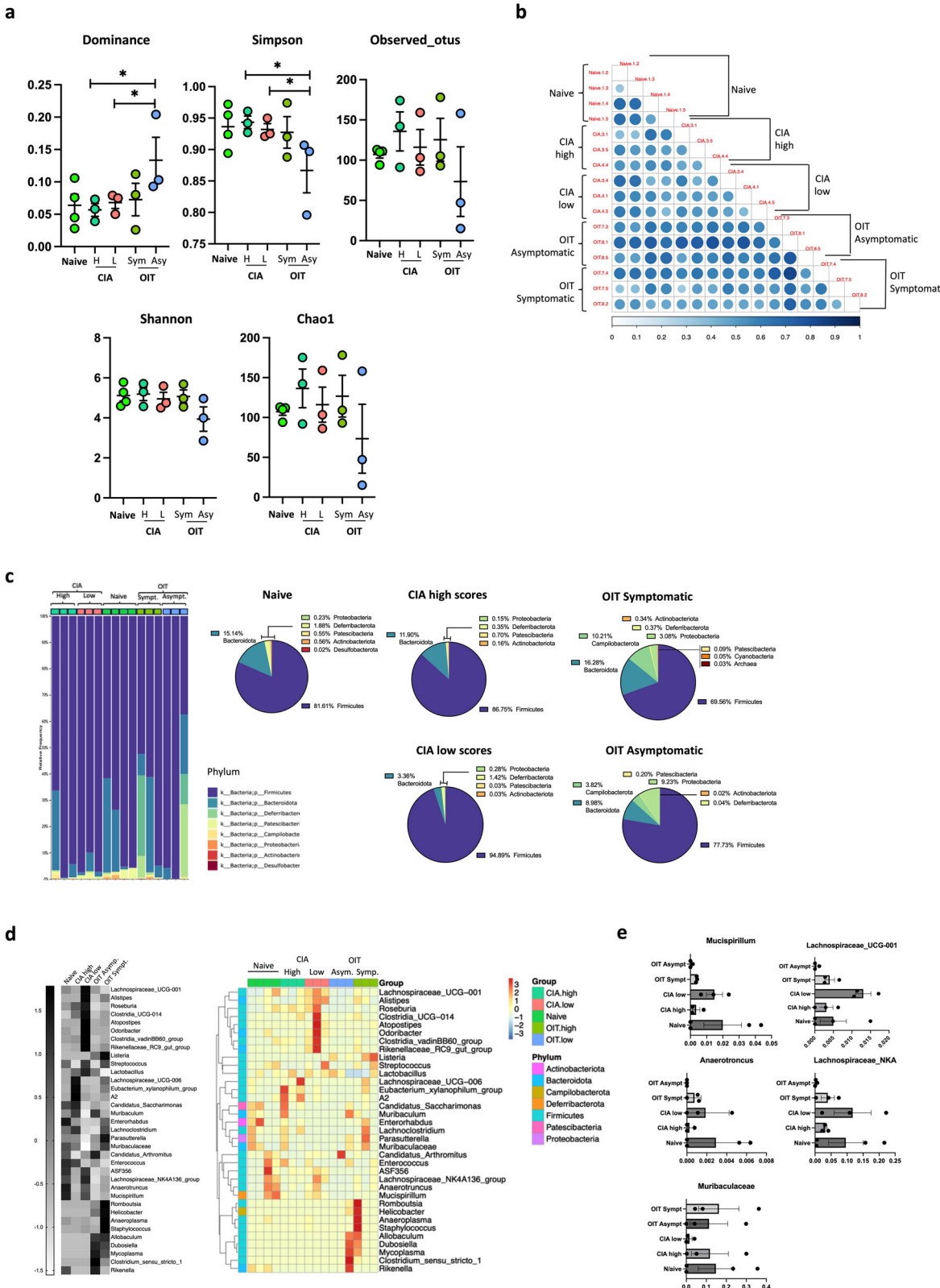

pathogens in the gut, but unresolved IL-17-associated inflammation may cause gut pathology[65]. In addition, IL-22 has been proposed to play both pathologic and protective roles in gut integrity and homeostasis. Thus, to identify pathological mechanisms, we examined the cells producing IL-17 and IL-22 in MLNs, ileum and colon as sites of Th17/22 induction and tolerance. Interestingly, whilst our results show upregulation of IL-17 in the

gut tissue from CIA but not asymptomatic OIT mice, the most striking changes appear to reflect OIT-induced switches in the cellular networks of IL-17[+] and IL-22[+] cells in the gut, with these cytokines being predominantly produced by innate effector cells like NKT and ILCs, rather than the CD4[+] helper and γδ T cells driving systemic autoimmunity in CIA[29]. IL-22 production was increased in all cell types isolated from the ileum in response to

**Fig. 8 | Analysis of the microbial composition in the ileum.** Faecal matter was taken from the ileum of naïve mice (n = 4), severe arthritis (CIA high, n = 3, disease scores 11, 10 and 7), mild arthritis (CIA low, n = 3; disease scores 3, 4 and 4), symptomatic OIT mice (OIT high, n = 3; disease scores 5, 6 and 4) and asymptomatic OIT mice (OIT low, n = 3) at day 33 when DNA was isolated and subjected to 16 S ribosomal RNA (rRNA) amplification and sequencing. **a** Alpha Diversity analysis, including observed_ otus, shannon, simpson, chao1, dominance and pielou_e indices. Each dot represents individual mice; graphs show the mean ± SEM. Kruskal–Wallis test was used to analyse whether the differences in species diversity between groups were significant, *$p < 0.05$. **b** Beta diversity indices heatmap of unweighted unifrac distance Matrix. The size and colour of the circle in the square represent the differences in coefficient between the two samples. The larger the circle

is, the darker the corresponding colour is, indicating that the differences between the two samples are greater. **c** Relative abundance of the indicated phyla. Each column shows data from individual mice. Data are also presented as pie charts, presenting proportion values for each group as means. **d** Clustering of Species Abundance. The top 35 genera in abundance were clustered from the species and sample levels according to their abundance information in each sample. Heatmap in grey scale shows the mean value of all mice in the group for Z value of taxonomic relative abundance after standardisation. The coloured heatmap represents the values for individual mice: the x-axis represents the sample name, and the y-axis represents the function annotation. The cluster tree on the left side is the species cluster tree. **e** Relativ**e** abundance of the indicated genera. Data show mean ± SEM and dots represent individual mice.

---

undenatured type II collagen administration, and this reflected an expansion of IL-22⁺ γδT cells, NKT cells and CD4 T cells. Overall, our data shows that the protective role of gut IL-17/IL-22 likely depends on the relative contribution coming from innate and adaptive immune cells, although we have not investigated all the molecular mechanisms underlying these protective cytokine and cellular networks.

Future studies plan to focus on the ability of undenatured type II collagen to harness protective IL-22 actions in the mucosal epithelium[50]. IL-22 promotes epithelial fucosylation, which is crucial to maintaining barrier integrity and immunity[66]. Indeed, perhaps linking the gut-joint axis, fucose can play a protective role in both local gut and systemic inflammation[67]. Interestingly, OIT restored the reduced gut fucosylation observed in CIA mice. We did not identify the mechanism(s) controlling such tissue fucosylation, but since this was not associated with changes in the expression of fucosyltransferases, environmental factors could be controlling local fucose content. For example, our results indicate that OIT induces microbiome changes that could be responsible for the modulation of fucosylation status as well as some of the therapeutic anti-inflammatory effects. In line with this, rewiring of immune-glycan-microbiome networks could affect the production of protective metabolites such as short-chain fatty acids with known anti-inflammatory effects in models of autoimmunity[68,69]. Additionally, specific microbial species may play an integral role in the induction of the protective glycan-dependent mechanisms associated with OIT. For example, segmented filamentous bacteria (SFB) is a common gut resident that can induce differentiation of Th17⁺ CD4 T cells in arthritis[70,71], but also induce IL-22 production in the small intestine lamina propria to modulate fucosylation of the ileum[72,73]. Thus, understanding the gut microenvironment, including cytokine networks, regulation of microbiome composition, and mechanisms linking both sides, such as epithelial glycosylation, may fill some of the gaps in the field, opening new avenues for translational opportunities, which consider human disease heterogeneity. For example, several clinical subgroups have been described in RA based on the presence of specific autoantibodies and/or specific immune responses; where for instance, the myeloid RA phenotype is driven by TNF, contrary to lymphoid RA type that is IL-17/IL-1-mediated, and these RA phenotypes correlate with response to treatments[74]. Thus, it is likely that specific groups of patients (with distinct pathogenic signatures) respond differently to undenatured type II collagen therapies, and the search for predictive biomarkers should be considered in future clinical trials. Additionally, patients' responses may differ in early or late disease stages, since anti-collagen antibodies show a higher concentration during early disease stages, and it has been associated with specific disease phenotypes[75,76].

Collectively, our study demonstrates the importance of differential IL-17- and IL-22-producing cell networks in the joint-gut inflammatory axis and the therapeutic potential of undenatured type II collagen. Regulation of cytokine networks correlates with protection against arthritis, indicating that factors affecting them may determine effective OIT. For translational purposes, understanding the impact of exogenous factors in regulating these immune networks can offer a way to enhance the clinical value of oral immunomodulation, and to identify patient groups which may benefit from these interventions. In this regard, we have identified molecular and cellular parameters in the gastrointestinal tract correlating with low inflammatory

conditions, including regulation of cytokines, cellular networks, tissue glycosylation and microbial profiles. These findings could offer new opportunities to treat arthritis and other chronic inflammatory disorders affected by changes in mucosal tissues.

## Methods
### Collagen-induced arthritis and undenatured type II collagen treatment
Male DBA/1 mice were purchased at 8–10 weeks of age (Envigo; Bicester, UK) and housed and maintained in the Central Research Facility of the University of Glasgow. We have complied with all relevant ethical regulations for animal use. All experiments were approved by and conducted in accordance with the Animal Welfare and Ethical Review Board of the University of Glasgow, UK Home Office Regulations and Licences PPL P8C60C865, PIL I62988261, PIL I675F0C46. To induce arthritis, mice were immunised with type II chicken collagen emulsion (1 mg/mL; in complete Freund's adjuvant [CFA] i.d.) and then administered type II chicken collagen in PBS i.p. at 21 days after the first injection. Mice were divided into three experimental groups, naïve, CIA, and mice receiving undenatured collagen. Undenatured type II collagen (7.33 mg/kg) was given to the animals by oral gavage three times a week, starting two weeks before induction of arthritis. The dose was selected based on pilot studies, suggesting that 7.33 mg/kg was the most protective under our lab experimental conditions. A concentrate of undenatured type II collagen was obtained from Lonza Greenwood LLC (UC-II® undenatured type II collagen). Monitoring of mouse health, clinical scores and weight were evaluated every 2 days. Clinical scoring of each limb was used to assess disease progression, where each paw received a score ranging from 0 (no inflammation) to 4 (highly inflamed/loss of functionality) as described previously[29]. An aggregated Clinical score of 10 for an individual mouse was considered an experimental endpoint, with any such mice immediately euthanized.

### Hematoxylin and eosin (H&E) staining
Gut tissue and paws from individual mice were collected for histology. Gut tissue was processed according to the Swiss roll technique, including duodenum, jejunum, ileum and colon. Gut tissue and paws were fixed in 4% paraformaldehyde for up to 24 h, then transferred into 70% ethanol for storage. Following dehydration, tissue was embedded in paraffin blocks and sectioned on a standard microtome at 7 μm thickness using a microtome (Leica RM2125). For joint blocks, bone decalcification, paraffin sectioning and H&E staining were completed at Histology Research Service, School of Veterinary Medicine, University of Glasgow. For gut tissue paraffin blocks, tissue processor Lecia Asap300 was used. Sections were heated for 35 min at 60 °C, then followed by H&E staining, which was performed on all tissues for identification of pathological changes. For analysis of gut sections, the length of gut villi and crypt depth were determined using Image J software to calculate villi/crypt length ratios. Pathology of the joint tissue (cell infiltration, pannus formation, bone damage and cartilage damage) was assessed by visual evaluation according to a score system ranging from 0 (no effect) to 4 (high pathology). The final score for each mouse paw is representative of two hind paws. No cut off was used to exclude any joints in the analysis.

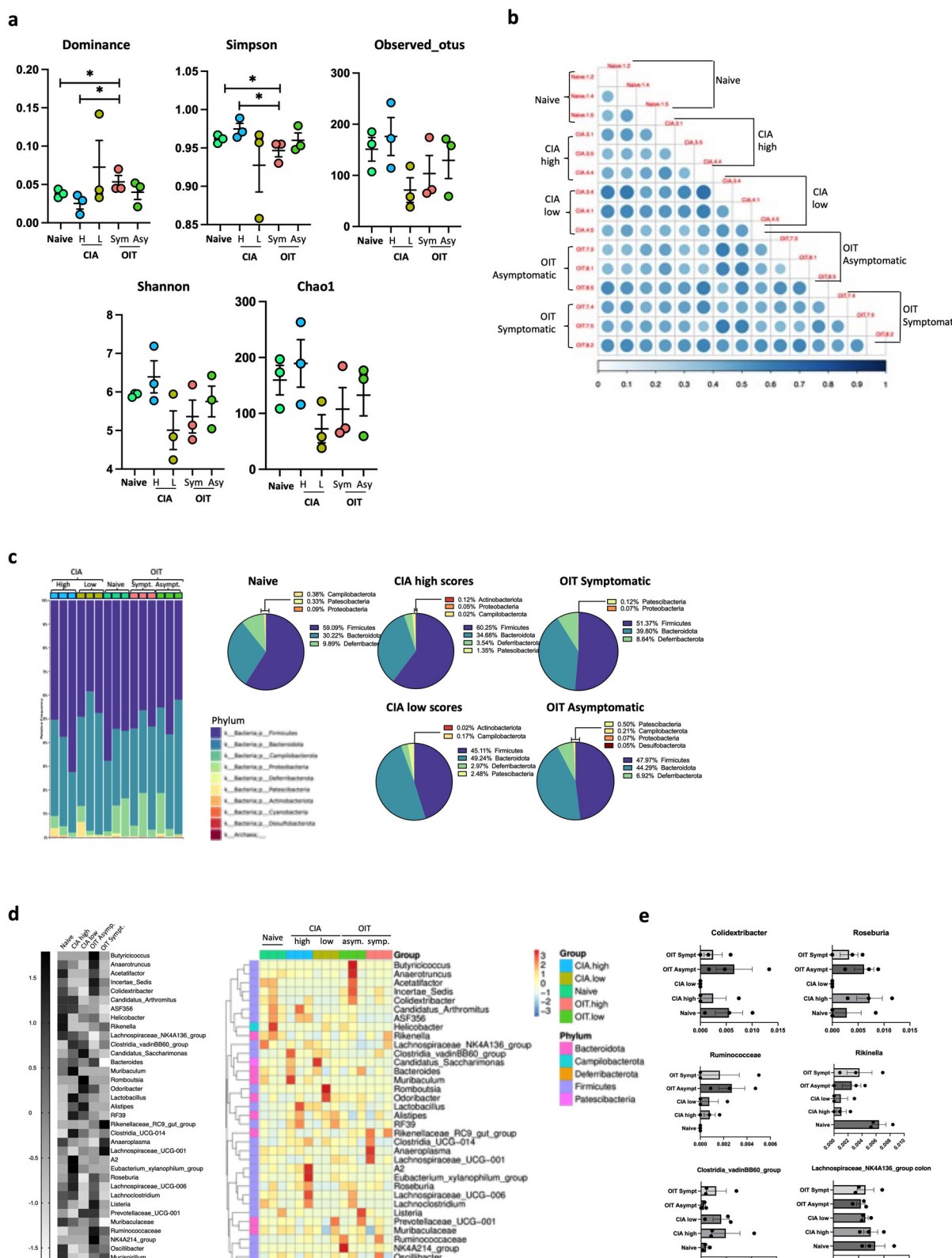

Assessment was conducted blindly, by two independent researchers. Images were acquired using an EVOS microscope.

**Periodic acid-Schiff (PAS) staining**
Paraffin sections were dewaxed using the same method used for H&E staining and rehydration. The slides were stained with Periodic acid for

5 min and rinsed with water before staining with Schiff's reagent for 15 min. After washing with tap water for 5 min, the sections were counterstained with haematoxylin for 1 min. Sections were then rinsed with water and dehydrated using a series of ethanol and xylene, mounted with DPX and coverslip. Images were acquired on an EVOS brightfield microscope.

**Fig. 9 | Analysis of the microbial composition in the colon.** Faecal matter was taken from the colon of naïve mice (*n* = 3), severe arthritis (CIA high, *n* = 3, disease scores 11, 10 and 7), mild arthritis (CIA low, *n* = 3; disease scores 3, 4 and 4), symptomatic OIT mice (OIT high, *n* = 3; disease scores 5, 6 and 4) and asymptomatic OIT mice (OIT low, *n* = 3) at day 33, when DNA was isolated and subjected to 16 S ribosomal RNA (rRNA) amplification and sequencing. **a** Alpha Diversity analysis, including observed_ otus, shannon, simpson, chao1, dominance and pielou_e indices. Each dot represents one individual mouse; graphs show the mean ± SEM. Kruskal–Wallis test was used to analyze whether the differences in species diversity between groups were significant, \**p* < 0.05. **b** Beta diversity indices heatmap of unweighted unifrac distance Matrix. The size and colour of the circle in the square represent the differences in coefficient between the two samples. The larger the circle is, the darker the corresponding colour is, indicating that the differences between the two samples are greater. **c** Relative abundance of the indicated phyla. Each column shows data from individual mice. Data are also presented as pie charts, presenting proportion values for each group as means. **d** Clustering of species abundance. The top 35 genera in abundance were clustered from the species and sample levels according to their abundance information in each sample. Heatmap in grey scale shows the mean value of all mice in the group for *Z* value of taxonomic relative abundance after standardisation. The coloured heatmap represents the values for individual mice: the *x*-axis represents the sample name, and the *y*-axis represents the function annotation. The cluster tree on the left side is the species cluster tree. **e** Relative abundance of the indicated genera. Data show mean ± SEM and dots represent individual mice.

### Immunofluorescence

Tissue sections (7 µm thickness) were dewaxed in xylene followed by 100% EtOH, 90% EtOH, 70% EtOH, 50% EtOH, and 30% EtOH for 3 min x2, respectively. Sections were then put in antigen retrieval buffer (Citrate buffer, Ph: 5.0) for 20 min at 95 °C[31]. To block non-specific antibody binding, sections were incubated with Carbo-Free Blocking Solution (Vector labs) for 30 min. The slides were washed in PBS-T (PBS 0.05% Tween20), and Streptavidin/Biotin blocking was performed following kit instructions (Vector laboratories, SP-2002). Sections were incubated with biotinylated lectins (Vector laboratories, 1:200) or primary antibodies in PBS and incubated overnight at 4 °C. Sections were washed with PBS-T at least 3 times. Streptavidin-Alexa 647 or secondary antibodies (1:400) were applied in PBS at room temperature for 60 min. Slides were rinsed with PBS and mounted with mounting media containing DAPI. Images were acquired with an EVOS microscope and confocal microscopy and were later analysed in Image J software.

### Serum cytokine and antibody ELISAs

Single-cell suspensions of lymph nodes (LN) were obtained, and cells were then cultured (10⁶/ml) and stimulated with TPA (50 ng/ml)/Ionomycin (500 ng/ml) for 12 h when supernatants were collected for detection of secreted cytokines. Interleukin-17 (IL-17) was measured by ELISA according to the manufacturer's instructions (Invitrogen). For determination of anti-collagen type II-specific IgG1 and IgG2a antibodies in serum, high-binding 96-well ELISA plates were coated with chicken collagen (5 µg/ml) overnight at 4 °C before washing and blocking with PBS 5% BSA. Serum from individual animals was diluted (1:100) and incubated with HRP-conjugated goat anti-mouse IgG1 or IgG2a (1:10,000) in PBS 10% FBS prior to developing with TMB. Samples were read in a microplate reader (Sunrise) at an optical density of 450 nm.

### Flow cytometry

LN cells were suspended in FACS buffer (1% FBS; 0.5 mM EDTA, in PBS). Lymphocytes were identified by labelling with antibodies against CD3 (FITC) and CD4 (AF700). CD8+ T cells were identified by labelling with antibodies against CD3 (FITC) and CD8 (PE-Cy5). B cells were identified by labelling with antibodies against CD19 (BV421). Data were acquired using an LSR-II Flow Cytometer, and populations were gated based on isotype controls using FlowJo software. For gut sample preparation, the ileum and colon were collected into HBSS 10% FBS following removal of all excess fat and Peyer's patches and immediately digested following established protocol in our group[77]. Briefly, tissue was washed with PBS and cut into small pieces (around 1 cm), rinsed with warm HBSS and transferred to wash buffer (HBSS no Ca⁺⁺, Mg⁺⁺ and containing 2 mM EDTA) for 15 min (37 °C, 220 rpm). Ileum samples were digested with 0.5 mg/mL of Collagenase IV (15 min, 37 °C) and the colon was digested with 0.5 mg/mL of Collagenase IV containing 24 µg/mL DNase I (20 min, 37 °C) to generate single cell suspensions prior to resuspending for antibody staining and flow cytometric analysis.

### RT-PCR

Gut tissues were dissolved in Trizol Reagent (1 ml/50–100 mg of tissue) following tissue homogenisation, and the lysates were centrifuged for 5 min at 12,000 rpm at 4 °C. The supernatants were transferred to a fresh tube, and chloroform (0.2 ml/mL of Trizol) was added to the samples for 5 minutes of incubation followed by centrifugation (15 minutes at 12,000*g* 4 °C). RNA in the aqueous phase was precipitated by adding 100% EtOH. Isolated RNA was cleaned using the RNeasy Plus Mini kit (Qiagen, Germany) according to the manufacturer's instructions. The High-Capacity cDNA Reverse Transcriptase kit (Applied Biosystems, Life Technology, UK) was used to generate cDNA. qPCR reactions were conducted using StepOne PlusTM real-time PCR system (Applied Biosystems, UK) and KiCqStart® qPCR Ready Mix (Sigma-Aldrich). Taqman probes were used to evaluate gene expression, including fucosyltransferases (FUT1, FUT2, FUT4, FUT7, FUT8 and FUT9). Data were normalised to the reference gene β-actin to obtain ΔCT values.

### Microbial diversity sequencing and analysis

Genomic DNA from the ileum and colon faecal matter was purified using QIAamp DNA Stool Mini Kit (Qiagen, Germany) and stored at −20 °C. All samples were subjected to paired-end sequencing by Novogene (UK) Company Limited, according to established protocols. Briefly, specific 16 S target hypervariable regions (amplicons) were amplified by PCR. Amplified DNA fragments were end-repaired, and A-tailed. The sequencing adapters were ligated to the ends of the DNA fragments using a DNA-binding enzyme, and the DNA fragments were purified using AMpure PB magnetic beads to construct an SMRTbell library. Finally, the sequencing primer was annealed to the SMRTbell templates, followed by the binding of the sequence polymerase to the annealed template. The library was checked with Qubit for quantification. Quantified libraries were pooled and sequenced on PacBio Sequel II/IIe systems. For sequencing data processing, the PacBio BAM file was split according to barcode and filtered to get clean data, used to generate Amplicon Sequence Variants (ASVs). Denoising, species annotation and alpha diversity indices (including observed_ otus, shannon, simpson, chao1, goods_ coverage, dominance and pielou_e indices), were performed by the DADA2 and Qiime2 software. Sequences with less than 5 abundance were filtered out to obtain the final ASVs and feature table. Then, the Classify-sklearn moduler in QIIME2 software was used to compare ASVs with the database and to obtain the species annotation of each ASV.

### Statistics

All data were analysed using one-way ANOVA with Fisher's LSD post-tests for parametric data or Kruskal–Wallis test and Dunn's post-test for non-parametric data (GraphPad Prism software). Shapiro–Wilk test was used to assess normality.

### Reporting summary

Further information on research design is available in the Nature Portfolio Reporting Summary linked to this article.

**Article**

## Data availability

Metagenomic data have been deposited in NCBI's Gene Expression Omnibus SRA under the Bioproject accession code PRJNA1117859. The source data behind the graphs in the paper can be found in Supplementary Data 1. All other data are available in the article and Supplementary files or from the corresponding authors upon reasonable request. Numerical source data for all graphs in the manuscript can be found in Supplementary Data 1 File.

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

## Acknowledgements

This work was funded by an Industrial partnership PhD between the University of Glasgow and Lonza and a Versus Arthritis Career Development Fellowship Award grant 21221.

## Author contributions

P.P., M.M.H., S.M. and M.P. conceived and designed the study. P.P., Y.W., M.H.N. and M.P. performed the lab work. P.P., Z.S., E.K.A., A.S., S.M., M.M.H. and M.P. contributed to data interpretation and manuscript revision.

P.P., S.M., M.M.H and M.P. wrote the paper. All authors read and approved the submitted version.

## Competing interests
Z.S., E.K.A. and A.S. are employees of Lonza Greenwood LLC., Greenwood, SC, USA. The remaining authors declare no competing interests.
