## [Peer Review File · Communications Biology]

Reviewers' comments:

Reviewer #1 (Remarks to the Author):

This manuscript by Pan et al describes the consequences of prophylactic oral administration of undenatured type II collagen on the incidence and severity of collagen induced arthritis, examining effects on joint pathology, immune cell subsets within draining lymph nodes and changes within the gut. The work describes a significant positive impact of oral collagen on disease incidence, paw swelling and histopathological changes. Attempts to address the mechanism underlying this protective effect reveal downregulation of IL17 (and to some extent IL22) in the joint (IF, but needs quantification) and draining lymph nodes. IL22 was up-regulated in the colon of the OIT group which prompted exploration of epithelial fucosylation – an important mechanism for maintenance of barrier integrity and immunity. Results showed restoration of reduced gut fucosylation in the setting of CIA. Finally, the authors explore the impact of oral denatured collagen on the microbiota suggesting it has a protective effect against arthritis induced changes.

Overall, this manuscript contains a significant amount of novel data and its exploration of the mechanisms by which oral collagen has a protective effect in arthritis adds appreciably to the field. I have questions around the way the data is presented (combining asymptomatic and symptomatic mice in the OIT group) and when addressed, this should give a better idea of the whether OIT impacts on just disease incidence or both incidence and severity of disease.

Major comments

1. The influence of oral undenatured collagen on disease incidence was very striking. Whilst in the CIA group (n=15) there was ~ 100% incidence, in the OIT group (n=16) incidence was reduced to ~55%. The authors need to determine/make clear whether further experiments reporting changes in disease severity, immunological status, gut pathology etc are a consequence of this reduced incidence. For example, in the 55% of OIT animals which did develop symptoms, was the disease less severe and gut changes less pronounced? Data needs to be presented comparing both the asymptomatic OIT animals and the symptomatic OIT animals back to the CIA group. In figure 1 – I would like to see graphs of the paw swelling and weight change with data for the asymptomatic and symptomatic OIT groups.
2. Following on from this point, there needs to be more clarity in the methods. For example, in Figure 1C/D – how many paws from each mouse were used for histology? Was there a cut off (e.g. only paws scoring 3 or 4 were used)? If an OIT animal was asymptomatic was histological assessment still performed on an uninflamed paw. By including asymptomatic OIT animals, it makes the disease look less severe, but was this really the case in OIT animals which did develop symptoms?
3. In figure 2B were draining lymph nodes harvested and analysed from asymptomatic mice – again this will influence the outcome and how the results are interpreted.

4. Figure 3A – I could not tell how this data was generated? Flow cytometry? Histology? Are both CIA and OIT animals included in the correlation analysis or just CIA animals – this needs further explanation of the methods in the main text.

5. Figure 7E – the analysis of the microbiome using QPCR – was this on gut tissue rather than mucus or faeces? With an n of two in one experimental group, I think it's very hard to draw meaningful conclusions from this data and it should be removed or extra animals added to the analysis. Further in the discussion it is stated that “the protective effect of undenatured collagen ... rewires cytokine networks and regulate the gut dysbiosis observed in arthritic mice”. Whilst I agree that structural changes in the gut were less severe in the OIT group, I don't think conclusions can be drawn about the microbiome.

6. I'm curious to understand how many of these changes in the gut would occur with administration of oral undenatured collagen in the absence of CIA. For example, do the authors believe that B cells in the MLN expand in response to oral collagen?

7. In the discussion, I would like to see some discussion of whether factors derived from inflamed joints cause damage to the gut and thus the reported reduced joint pathology in the OIT group may be impacting on the gut damage.

8. Figure 3C – The authors state in the results that “ IL17 and IL22 were up-regulated in the joint from CIA but not OIT mice” in Figure 3C one IF image is shown for each cytokine/treatment – is this representative of a higher n number? Is it possible to quantify this to support their statement?

Minor comments

1. Can the authors show the gating strategy for ILC3, gamma delta T cells and NKT cells please?

2. Figure 7, panels C and D should be labelled with ileum and colon

3. Figure 7D the legend describes a one way ANOVA, what was the post-hoc test?

4. In Figure 2b, total numbers of cells in 3 different draining lymph nodes were quantified – were all three lymph nodes collected irrespective of which paws were inflamed?

5. In reference to figure 3D the authors say “undenatured collage reduced IL17 producing cells in DLN cell from OIT mice...” given that this is results from an ELISA this should be changed to “reduced production of IL17” as it may be the same number of cells, but they are producing less.

6. Can sequences for 16s rRNA QPCR primers/probes be provided?

7. Why was chicken collagen used to induce arthritis in the DBA/1 mice – does oral undenatured collagen not protect against CIA induced with bovine collagen (more standardly used in this strain)?

8. Typo “Thus, it is likely that specific groups of patients (...) response” - response should be respond.

9. Why were only Fut1, 2 and 8 chosen to quantify by QPCR?

Reviewer #2 (Remarks to the Author):

This study examines the impact of orally administering undenatured type II collagen on gut and joint inflammation in a Collagen-Induced Arthritis (CIA) experimental model. The authors leverage the phenomenon of oral tolerance, which can suppress inflammatory immune responses. They suggest that collagen treatment offers protection against gut pathology by influencing the IL-17/IL-22 pathways and associated cellular networks, as supported by the data. However, the precise mechanisms through which oral tolerance alleviates gut inflammation remain partially elucidated.

Here are some of my concerns

1. There seems to be a mix-up in Fig 1A, where the plots of the clinic score and paw width appear to be switched.

2. The disease incidence depicted in Fig. 1B lacks statistical analysis, which should be included to support the presented findings.

3. To enhance the clarity of the immunostaining of IL-17 and IL-22, high-magnification images are needed. In Fig 6, the IL-22 staining appears to be concentrated in the epithelium, but this observation should be verified. Additionally, if the staining is indeed accurate, it raises questions about the authors' focus on immune cells for their analysis of cytokine expression, despite the prominent presence of IL-22 in enterocytes.

4. Figure 7 does not evaluate the overall impact of CIA and OIT on intestinal microbes, and it would be beneficial to include this analysis to provide a comprehensive understanding of the study's findings.

Reviewers' comments:

Reviewer #1 (Remarks to the Author):

This manuscript by Pan et al describes the consequences of prophylactic oral administration of undenatured type II collagen on the incidence and severity of collagen induced arthritis, examining effects on joint pathology, immune cell subsets within draining lymph nodes and changes within the gut. The work describes a significant positive impact of oral collagen on disease incidence, paw swelling and histopathological changes. Attempts to address the mechanism underlying this protective effect reveal downregulation of IL17 (and to some extent IL22) in the joint (IF, but needs quantification) and draining lymph nodes. IL22 was up-regulated in the colon of the OIT group which prompted exploration of epithelial fucosylation – an important mechanism for maintenance of barrier integrity and immunity. Results showed restoration of reduced gut fucosylation in the setting of CIA. Finally, the authors explore the impact of oral denatured collagen on the microbiota suggesting it has a protective effect against arthritis induced changes.

Overall, this manuscript contains a significant amount of novel data and its exploration of the mechanisms by which oral collagen has a protective effect in arthritis adds appreciably to the field. I have questions around the way the data is presented (combining asymptomatic and symptomatic mice in the OIT group) and when addressed, this should give a better idea of the whether OIT impacts on just disease incidence or both incidence and severity of disease.

We would like to thank the reviewer for taking the time to review our manuscript and for providing such constructive comments. We have conducted additional experiments and reanalysed our data following their suggestions, to show both asymptomatic and symptomatic OIT mice in our data analysis. This was a very useful suggestion that has helped us to further understand the mechanisms triggered in our experimental model and improve the manuscript.

These new results in terms of analysis are presented in the revised version of the manuscript, as detailed in the point-to-point reply.

Major comments

1. The influence of oral undenatured collagen on disease incidence was very striking. Whilst in the CIA group (n=15) there was ~ 100% incidence, in the OIT group (n=16) incidence was reduced to ~55%. The authors need to determine/make clear whether further experiments reporting changes in disease severity, immunological status, gut pathology etc are a consequence of this reduced incidence. For example, in the 55% of OIT animals which did develop symptoms, was the disease less severe and gut changes less pronounced? Data needs to be presented comparing both the asymptomatic OIT animals and the symptomatic OIT animals back to the CIA group. In figure 1 – I would like to see graphs of the paw swelling and weight change with data for the asymptomatic and symptomatic OIT groups.

We have addressed this point by separating the OIT group into symptomatic and asymptomatic groups as suggested. In fact, this was a very useful suggestion as it has helped us to understand and interpret our results as well as to plan and execute further experiments to support our hypotheses.

Specifically, the revised version of the manuscript presents clinical scores and incidence in Figure 1A, and Figure 1B now shows the paw swelling and weight change for the naïve, CIA and asymptomatic and symptomatic OIT mice. The latter parameters in symptomatic OIT mice were not significantly different to those in the CIA group, whilst asymptomatic mice resembled healthy mice both in terms of paw swelling and weight change. In figure 1C-D for histology and joint tissue pathology, the data were also separated in this way and showed that OIT effectively prevented cartilage and bone damage in all asymptomatic mice. Overall, this led us to conclude that OIT is predominantly affecting incidence of joint destruction rather than disease severity, an important point that is now highlighted in the Results and Discussion sections.

To clarify the immunological status of both symptomatic and asymptomatic mice, we have focused the first half of the manuscript on comparing both groups with naïve and CIA control mice. We have conducted new

experiments to have statistically-relevant numbers of mice in both OIT groups, and the manuscript now shows such analysis in terms of:

- Anti-collagen antibodies, IgG1 and IgG2a, Figure 2A.
- Immune populations in draining lymph nodes, Figure 2B-C.
- IL-17 and IL-22 in immune cell populations in draining lymph nodes, Figure 3.
- Gut damage and immune cell populations in mesenteric lymph nodes, Figure 4.

Overall these new data revealed that asymptomatic OIT mice exhibited reduced IL-17 and IL-22 expression in the joints, whereas symptomatic OIT mice did not achieve this, showing cytokine expression at comparative levels to those of CIA mice. However, the antibodies produced against collagen did not show any relevant differential expression between the two OIT groups, supporting the importance of cellular rather than humoral responses in OIT protection. Furthermore, analysis of the gut pathology at cull during the period of the established phase of disease showed that overall neither the symptomatic and asymptomatic OIT mice had the structural gut damage associated with arthritis, although there was some evidence of disruption of the villi/crypt length ratio in the colon, for asymptomatic mice. We believe that this could be a consequence of the dual pro-inflammatory and tissue protective roles described for IL-22, for example when it can be protective in the gut tissue by promoting epithelial regeneration. This has been described, referenced, and discussed in the revised version of the manuscript.

In the next part of the manuscript, encompassing figures 5-7, we show results from the OIT asymptomatic cohort compared with naïve and CIA mice. These experiments are highly labour intensive, given the technical challenges associated with gut tissue digestion and cell isolation. Thus, working with both symptomatic and asymptomatic OIT mice would have compromised the integrity of the cells during isolation and we focused on the identification of the protective mechanisms triggered in those mice that achieved protection against disease, with particular attention to the gut tissue. To achieve this goal, we have removed all the symptomatic mice from these data sets, leaving only asymptomatic mice in the results. Therefore, our revised data shows the IL-17, IL-22 and gut fucosylation rewiring associated with protection against arthritis upon effective OIT.

Finally, we have conducted a new study characterising the gut microbiome composition using 16S rRNA high-throughput sequencing to replace the PCR data. For this, we analysed the microbiomes of symptomatic and asymptomatic OIT mice separately and the results are detailed in our response to point 5.

2. Following on from this point, there needs to be more clarity in the methods. For example, in Figure 1C/D – how many paws from each mouse were used for histology? Was there a cut off (e.g. only paws scoring 3 or 4 were used)? If an OIT animal was asymptomatic was histological assessment still performed on an uninfamed paw. By including asymptomatic OIT animals, it makes the disease look less severe, but was this really the case in OIT animals which did develop symptoms?

We have clarified the methods to provide more experimental detail. For histology, we have indicated that individual hind paws were analysed. All paws were examined in a double blinded way, so all paws were scored, including the non-inflamed paws. This has now been clarified in the methods section and figure legends. A similar approach has been taken to review and clarify the rest of the methods and figures.

3. In figure 2B were draining lymph nodes harvested and analysed from asymptomatic mice – again this will influence the outcome and how the results are interpreted.

Please, see reply to point 1 where this has been explained in detail. In summary, we have now separated all data in figures 1 to 4 into symptomatic and asymptomatic OIT mice when possible.

4. Figure 3A – I could not tell how this data was generated? Flow cytometry? Histology? Are both CIA and OIT animals included in the correlation analysis or just CIA animals – this needs further explanation of the methods in the main text.

Apologies for the confusion in presenting these results. Our aim in this experiment was to evaluate whether IL-17 and IL-22 expression correlated with disease severity, since IL-22 has been sometimes described as a cytokine with dual effects in inflammatory conditions, including joint disease in the CIA model (Pineda et al., 2014). Therefore, we analysed all groups to establish correlations between cytokine expression and disease scores using all groups. These data were generated by flow cytometry, and the results supported the pro-inflammatory activity of these cytokines in our model. We have now amended the figure, which confirms the pathogenic role of IL-17⁺ and IL-22⁺ CD4 T cells. This has been also clarified in the text:

'To confirm the pathogenic role of IL-17 and IL-22 in the CIA joint, we first collated mice from all groups to directly compare their clinical scores with IL-17 and IL-22 expression in CD4⁺ T cells (Figure 3A) and B cells (Figure 3B), analysed by flow cytometry. Results indicate that levels of IL-17⁺ and IL-22⁺ CD4⁺ T cells significantly correlate with disease severity, regardless of treatment'

5. Figure 7E – the analysis of the microbiome using QPCR – was this on gut tissue rather than mucus or faeces? With an n of two in one experimental group, I think it's very hard to draw meaningful conclusions from this data and it should be removed or extra animals added to the analysis. Further in the discussion it is stated that "the protective effect of undenatured collagen ... rewires cytokine networks and regulate the gut dysbiosis observed in arthritic mice". Whilst I agree that structural changes in the gut were less severe in the OIT group, I don't think conclusions can be drawn about the microbiome.

We agree with the reviewer that this experiment needed more power to deliver meaningful data. Therefore, we decided to conduct 16S rRNA high-throughput sequencing in DNA extracted from faecal content obtained from ileum and colon. We have analysed 4 naïve mice, 6 CIA mice and 6 OIT mice. For analysis, OIT mice were separated into symptomatic and asymptomatic, and CIA mice were separated into high disease scores (~ 10) and lower, but still severe scores (~4).

As the reviewer comments, the structural changes in the gut of both groups of OIT mice looked less severe overall than those of CIA mice. Our new data set show that asymptomatic OIT mice have a distinct microbiome, both in ileum and colon to that of Naïve and CIA mice. We think that these differences are associated with the different cytokine networks and fucosylation profile found in these groups, although further work needs to be done to gain insight into potential mechanisms connecting these factors. Nevertheless, our data provide support for formulating new hypotheses in this regard, that we will follow up in future studies. These data have been presented in the results and commented on in the discussion.

Interestingly, we observed that CIA mice with higher disease scores exhibited a microbiome composition closer to that of the naïve group than those with lower disease scores, and that this composition was potentially less pathogenic based on the alpha indexes. However, it is likely that mice with high disease scores (and that have had a longer disease) may be entering into the resolving phase of the CIA model (Marinova-Mutafchieva et al., Clin Exp Immunol 2006), a phase not observed in human RA. Therefore, such analysis provides relevant results that may help to plan and understand results generated in experimental models for translation to human disease. Interestingly, we have previously shown that microbiome dysbiosis occurs early, during initiation of CIA, preceding challenge with collagen and onset of joint disease (Doonan et al., Nature Communications, 2019). Collectively, in conjunction with drawing our conclusions on how to identify protective microbiome changes in OIT mice by using the early arthritic group (CIA-low) as a reference, these findings may suggest that restoration towards microbiome homeostasis may be an early mechanism leading to resolution of joint disease, as levels of inflammatory IL-17 (a key inflammatory driver in the model) were still comparable between mice with CIA high and low scores (Rebuttal Figure 1) during this time period of the model.

Rebuttal Figure 1: Number of IL-17⁺ draining lymph nodes cells. Data generated by intracellular cytokine staining and Flow Cytometry in total cells.

6. I'm curious to understand how many of these changes in the gut would occur with administration of oral undenatured collagen in the absence of CIA. For example, do the authors believe that B cells in the MLN expand in response to oral collagen?

This is a relevant question, but we have not been able to address it yet. Our data show: i) levels of anti-collagen antibodies in serum are quite similar in CIA and OIT mice in our model; ii) some expansion of B cells in the MLNs of symptomatic OIT mice, and iii) some indication of an expanded population of B cells in the gut tissue of asymptomatic OIT mice. These data are obtained at a late disease stage and cannot provide an answer about early effects on B cells in the gut tissue or on naïve steady-state responses. This is a direction that we would like to explore in follow-up studies, including the study of Breg activity. We do believe that some B cells in the MLNs will be affected in response to oral collagen, something that would have an impact in the responses described in this manuscript. We have commented on this in the discussion.

7. In the discussion, I would like to see some discussion of whether factors derived from inflamed joints cause damage to the gut and thus the reported reduced joint pathology in the OIT group may be impacting on the gut damage.

This is a very interesting area of research, and a question that has no clear answer at present. Whether disease progression reflects gut to joint effects, or the joint to gut direction is unclear, although there is probably space for both cases among clinical cases. In any case, changes in the gut microbiome, and factors that control this microbiome, seem to be a necessary event to lead RA pathogenesis. Indeed, we and others have previously shown (Doonan et al., Nature Communications, 2019, Jubair et al., Arthritis Rheumatology, 2018) that microbiome dysbiosis takes place early in initiation of CIA, preceding systemic inflammation and joint disease.

This has been further clarified in the revised Discussion section, using IL-17 as an example of a modulated cytokine in our model. We have also discussed how IL-22 from the joint could play an opposite role, by inducing epithelial regeneration and gut protection.

8. Figure 3C – The authors state in the results that “ IL17 and IL22 were up-regulated in the joint from CIA but not OIT mice” in Figure 3C one IF image is shown for each cytokine/treatment – is this representative of a higher n number? Is it possible to quantify this to support their statement?

We have quantified the staining (n=3 for each group), observing a significant difference in IL-22 but not in IL-17 expression. We have modified this statement in the results section.

Minor comments

1. Can the authors show the gating strategy for ILC3, gamma delta T cells and NKT cells please?

All gating strategies have been added to the manuscript as supplementary figures.

2. Figure 7, panels C and D should be labelled with ileum and colon

We have labelled these.

3. Figure 7D the legend describes a one way ANOVA, what was the post-hoc test?

Post-hoc test used was Dunn's post-test. Please notice that results in Figure 7 include only asymptomatic OIT mice, as explained in previous points.

We have added a statement for statistical analysis in the methods section: *'All data were analysed using one-way ANOVA with Fishers LSD post-tests for parametric data or Kruskal–Wallis test and Dunn's post-test for non-parametric data (GraphPad Prism software). Shapiro-Wilk test was used to assess normality.'*

4. In Figure 2b, total numbers of cells in 3 different draining lymph nodes were quantified – were all three lymph nodes collected irrespective of which paws were inflamed?

The data shown in figure 2 about draining lymph nodes are from all mice, regardless of the inflammation. This has been indicated in the revised version of the manuscript. However, the new analysis separating symptomatic and asymptomatic OIT mice has a more uniform distribution, and clinical scores are more consistent compared with the original analysis, improving the interpretation of data and subsequent analysis.

5. In reference to figure 3D the authors say “undenatured collage reduced IL17 producing cells in DLN cell from OIT mice...” given that this is results from an ELISA this should be changed to “reduced production of IL17” as it may be the same number of cells, but they are producing less.

This has been changed.

6. Can sequences for 16s rRNA QPCR primers/probes be provided?

We have replaced the qPCR data with 16S rRNA high-throughput sequencing to identify changes in gut microbiome composition across all the treatment groups investigated.

7. Why was chicken collagen used to induce arthritis in the DBA/1 mice – does oral undenatured collagen not protect against CIA induced with bovine collagen (more standardly used in this strain)?

Undenatured collagen would also protect against CIA induced with bovine collagen, differences between chicken and bovine collagen in the process of breaking immune tolerance during disease initiation in mice being minimal. We used chicken collagen to keep a more defined experimental setup, since the orally administered collagen was produced from chicken tissue. Furthermore, using chicken collagen in the CIA model usually provides a higher disease incidence and severity score than that of bovine collagen.

8. Typo “Thus, it is likely that specific groups of patients (...) response” - response should be respond.

Thanks, this was fixed.

9. Why were only Fut1, 2 and 8 chosen to quantify by QPCR?

We analysed these because we identified them as the most expressed ones in the tissue, but the revised version of the manuscript now additionally includes expression of Fut4, 7 and 9.

Reviewer #2 (Remarks to the Author):

This study examines the impact of orally administering undenatured type II collagen on gut and joint inflammation in a Collagen-Induced Arthritis (CIA) experimental model. The authors leverage the phenomenon of oral tolerance, which can suppress inflammatory immune responses. They suggest that collagen treatment offers protection against gut pathology by influencing the IL-17/IL-22 pathways and associated cellular networks, as supported by the data. However, the precise mechanisms through which oral tolerance alleviates gut inflammation remain partially elucidated.

Here are some of my concerns

1. There seems to be a mix-up in Fig 1A, where the plots of the clinic score and paw width appear to be switched.

Thanks, we have corrected this.

2.The disease incidence depicted in Fig. 1B lacks statistical analysis, which should be included to support the presented findings.

The graph with disease incidence shows no stats because the statistical significance of the results was already assessed in the disease scores data and paw width, both datasets including individual mice. Please notice that disease incidence is a cumulative value expressed in percentage, which means that there is one value per day and group. However, quantification of the area under of curve for CIA and OIT mice, shows that both groups are different [CIA, 1071 and OIT, 763.7]. We believe that the statistical significance of the OIT effect is better represented in the disease score and paw width data, but the analysis of the area under the curve can be included in the manuscript if the reviewer considers it essential.

3.To enhance the clarity of the immunostaining of IL-17 and IL-22, high-magnification images are needed. In Fig 6, the IL-22 staining appears to be concentrated in the epithelium, but this observation should be verified. Additionally, if the staining is indeed accurate, it raises questions about the authors' focus on immune cells for their analysis of cytokine expression, despite the prominent presence of IL-22 in enterocytes.

Thanks for these suggestions, which provide us with an opportunity to clarify our results. We have now included new higher magnification images of staining for IL-22 in the gut tissue (Figure 6H) indicating that the reviewer is correct in suggesting that IL-22 staining shows higher intensity in epithelial areas. This clarification of the staining pattern has been described in the text of the revised manuscript, referring to this positive staining as '*IL-22 secretion/deposition in the tissue epithelium*' in the results section.

Regarding expression of IL-17 and IL-22 in joints and synovial tissue, this has been extensively described in the literature, including some of our previous work (Pineda et al., *Arthritis Rheum* 64, 3168-3178, doi:10.1002/art.34581 (2012) and Pineda et al., *Arthritis Rheumatol* 66, 1492-1503, doi:10.1002/art.38392 (2014)), and we would direct the reviewer to these publications (both cited in the submitted manuscript). Therefore, we consider that the presented analysis provides sufficient quantitative power to compare their expression in joints amongst groups, which was the goal of these experiments.

The reviewer poses an interesting question regarding the expression of IL-22 by immune system cells, and the presence of this cytokine in enterocytes. We acknowledge the reviewer's comment, and we think that the staining of IL-22 in enterocytes does not contradict its expression by immune system cells, but rather complements it. IL-22 has often been described as a cytokine connecting the immune and stromal compartments, since it is synthesised by adaptive and innate immune system cells (T cells and innate-like lymphocytes, like ILCs, NKs or gamma-delta T cells), although the IL-22R1 subunit is not typically expressed in immune system cells but it is found in epithelial cells in gastrointestinal tract. Thus, by analysing production by immune system cells, in conjunction with this new detection of staining of enterocytes, we have a broader picture of IL-22 function by defining both its origin and its tissue targets in this model. As described in the text, we consider that IL-22 staining in enterocytes corresponds with cytokine deposition, whereas flow cytometry analysis reveals cytokine production. These data fit with the described role of IL-22, since interaction between IL-22 and its receptor is involved in regulating the immune response at mucosal surfaces and promoting tissue protection and repair. This, in relation to our new data in OIT mice has now been extensively described and discussed in the revised manuscript.

4.Figure 7 does not evaluate the overall impact of CIA and OIT on intestinal microbes, and it would be beneficial to include this analysis to provide a comprehensive understanding of the study's findings.

We agree with the reviewer that the results in figure 7 did not evaluate the overall impact of CIA and OIT on intestinal microbes. To achieve this goal, we have conducted new 16S rRNA high-throughput sequencing in DNA extracted from faecal content obtained from ileum and colon faecal samples. These data are now presented in the revised manuscript in Figure 8 (ileum microbiome), Figure 9 (colon microbiome) and Supplementary Figure 7 (prediction of metagenomic function). These data include 4 naïve mice, 6 CIA mice, and 6 OIT mice, the latter two separated into two subgroups based on their inflammatory profile (see also comments to reviewer 1, point 5).

Reviewer 3

RA is a chronic, autoimmune disease affecting the synovial joints. In this paper, the authors used a model of Oral Immune Therapy (OIT), and treated CIA mice with undenatured collagen. They reported that the alleviated effect of undenatured collagen was due to the regulation of IL-17 and IL-22 cytokines in the joint and rewiring of associated gd T cell and ILC cytokine networks in the gut. This study is interesting, however, there are several major concerns for data interpretation and conclusion.

1. It is necessary to show the data in bar graph about the effect of OIT on CD4+ T cell number or frequency besides the radar charts in Fig 3G.

We have represented all the data depicted in the radar charts as bar graphs in the Supplementary Figures section. This addition complements the value of the radar charts, which offer a comprehensive overview of all evaluated parameters and trends.

2. In this study, authors indicate the OIT therapy may exert the benefit by regulating IL-17 and IL-22, they also showed the several cell types contributing to IL-17 and IL-22. Whether intestinal IL-22 is protective or pathogenic is still unclear, therefore, the conclusion is not convinced.

Thanks for this comment. In response to Reviewer 1's feedback, we have categorized OIT mice into symptomatic and asymptomatic groups, helping us further understand the role of intestinal IL-22 during OIT and inflammatory arthritis. The question of whether IL-22 is protective or pathogenic in arthritis is highly relevant and has been extensively explored by various research groups. In fact, the answer may not be straightforward, as we and others have demonstrated that IL-22 exhibits dual actions in inflammatory conditions including the CIA model (Geboes, *Arthritis Rheum* **60**, 390-395, 2009; Pineda, M. A., *Arthritis Rheumatol* **66**, 1492-1503; Justa, S. *PLoS One* **9**, 2014) acting as either pro- or anti-inflammatory depending on the context, local environments and stage of disease.

While providing a definitive answer to this biological question therefore exceeds the scope of our manuscript, our data offer valuable insights into understanding IL-22's actions. Our new data show that all OIT mice exhibit protected gut tissue, even those with high joint inflammatory scores, and this correlates with increased IL-22 expression in the gut and rewired IL-22 networks. Therefore, although we did not demonstrate direct causality, our results strongly support the hypothesis of IL-22 being protective in gut tissue in CIA. This aligns with the described regenerative actions of IL-22 in the gut, a point that has been discussed in the revised manuscript. Furthermore, we provided a broader description of potentially intertwined IL-22 mediated pathways that may affect local and systemic inflammation, such as gut fucosylation and microbiome changes. The possible mechanisms affected by these networks, and potential therapeutic opportunities, are discussed in the manuscript.

Overall, we think that our data establish the groundwork for further mechanistic studies to evaluate the mechanisms triggered by IL-22, something that we will further investigate in more focused studies. Please, see the reply to Review 1 point 1 for additional discussion of this topic.

3. Authors showed that OIT can also alter gut glycosylation. Although in healthy gut, the support of IL-22 on homeostatic expression of mucins is associated with fucosylation, it is unclear whether fucosylation is necessary in maintain gut integrity and in CIA alleviation.

Whether IL-22 is essential to maintain gut integrity in OIT mice is a question that is directly related to the previous comment. The reviewer correctly states that IL-22 induces fucosylation of mucins, leading to increases in glycotransferases Fut2, Mgat4a, and Mgat4b, which ultimately regulates interactions with intestinal microbes to prevent colonization of pathogenic bacteria (Daniela J. Carroll, *Journal Bio. Chemistry* 2022; Pickard J.M. *Nature*. 2014; Pham T.A; *Cell Host Microbe*. 2014; Nagao-Kitamoto H. *Nat. Medicine* 2020). We show that OIT mice have increased gut fucosylation, providing further support of our hypothesis of OIT inducing protective IL-22-fucosylation-microbiome networks. We have not provided direct evidence of a direct link between fucose and gut integrity, although this is supported by published work in microbiology and inflammation. Moreover,

we have identified bacterial genera that are expanded in OIT asymptomatic mice, such as Ruminococcaceae and Roseburia that have been described as producers of protective metabolites that are also involved in promoting local IL-22 production.

Thus, like in the previous point, our goal with these experiments was to provide a broad and comprehensive description of the gut changes associated with protection, as the overall picture highlights the involvement of multiple factors. We now plan to use this data to further explain mechanisms focusing on the specific areas indicated by these studies, for example addressing in future work the functional role of fucose in maintaining gut integrity and CIA amelioration.

4. Authors investigated the gut microbiota composition, while it is better to explain and interpret these data since there are many papers showing the GM with gut integrity or immune cells regulation in the gut. Since authors indicated the immune cells change in the gut, gut integrity, they should also discuss these relationship in the paper.

The reviewer indicates that immune system cells in the gut tissue modulate the composition of the gut microbiota, highlighting that the cellular changes that we observe during OIT may be playing similar roles in our model. This is a very relevant point, since maintaining the right interplay between immune system cells and the gut microbiota is critical to preserve health and tissue homeostasis.

As described in our responses to the other reviewers, we have conducted new 16S RNA sequencing of faecal ileum and colon samples to provide a broader perspective of protective and pathogenic changes in microbiota across all the experimental groups (See also reply to reviewer 1 question 5). Furthermore, we have focused the analysis of immune system cell populations to the asymptomatic OIT mice, in order to further understand the factors that could be affecting the crosstalk between microbiota and immune system cells in the context of arthritis and the protective induction of oral tolerance. These new data are now included in the revised paper (see also reply to reviewer 1, point 1) and their relevance has been commented on in the discussion.

5. Treg is usually detected as CD3+CD4+CD25+Foxp3+ in mice, what is the meaning to detect CD39+ or CD73+ Treg cells in this paper?

Thanks for this comment, which will improve the clarity of the manuscript. We initially investigated Tregs and Treg subsets, using the CD39 and CD73 markers as indicators of their functional suppressor activity, given the relevance of these cell populations in inducing oral tolerance. However, our data indicated that, at least at the cull day, they did not appear to play a leading role in protecting against arthritis. Nevertheless, it is conceivable that they may still be necessary to trigger immune mechanisms in the early stages of the disease. Our data ultimately led us to identify changes in effector cellular responses as the key mechanism underlying protection. We believe that, by relegating the Treg negative data to Supplementary Figures, we can direct attention more effectively to our main findings in the paper. Therefore, in the revised version of the manuscript, we have clarified the characterisation of the Treg subsets and moved these data to a Supplementary Figure to present a more focused analysis of our data.

Reviewers' comments:

Reviewer #1 (Remarks to the Author):

I am pleased to see that the authors have considered the difference between asymptomatic and symptomatic OIT mice, and I am happy that this has helped to draw conclusions about the differential effects on disease incidence and disease severity. I appreciate this was a significant amend to the presentation of the results. The manuscript now describes how OIT had a protective effect, and reduced the incidence of CIA. However, those OIT mice which did go on to develop symptoms exhibited the same degree of severity. Interestingly, the OIT prevented damage to the gut even in mice which went on to develop symptoms. To me this would suggest that breach of tolerance at mucosal surfaces does not have to occur in order to initiate joint disease, and would help with the cause and effect argument. The findings have been in the main clearly communicated in the updated version. For absolute clarity, could the authors add into the beginning of the results section that all (I think it was 100%) the mice in the CIA group developed symptoms. This would be helpful. There are a few other points in the text (described below) where the differences between experimental groups need explaining.

There is the implication that OIT somewhat reduced inflammatory responses in the joints of symptomatic mice – this seems to mostly hinge on data presented in Figure 3C/D. This figure needs to be clear on whether the OIT group refers to asymptomatic or symptomatic animals as this is a critical point.

The 16S rRNA seq adds further detail to the paper. There seems to be a striking reduction in diversity in the ileum of the asymptomatic OIT group in particular in the Shannon index [Shannon is misspelt in the figure itself and text] and the results section describes " lower diversity". Yet this contrasts with what is said later on "Overall, OIT increased the diversity of bacteria in the ileum, ..." Can this be modified? Additionally, the PICRUST2 analysis may need some further explaining. The pathways on the figure (SF7) are just labelled with the number. To say that OIT mice show a "distinct metabolite profile in the colon" is a bit misleading if this is an interpretation of figure S7B.

The authors have addressed all my other points and I thank them for these responses.

Additional comments

Abstract – "Incidence was reduced to 50%, correlating with down regulation of interleukin 17 and IL 22 in the joints" Does this statement still hold? Is it not referring to the previous version of the data?

Introduction "Current immunosuppressive treatment still shows 20-30% of non-responders" check for sense?

Introduction "Mechanistically, oral collagen administration promotes regulatory T cell function..." where? Can the authors state in the gut, joint or both?

Results – When the prophylactic undenatured type II collagen oral immunotherapy regime is first mentioned (line 5 of the results section) can the authors briefly explain that this starts 2 weeks prior to CIA induction and is 3x per week – for clarity.

Results – "To evaluate whether OIT...to measure paw swelling and relatively weight change" should be relative not relatively.

Results – "symptomatic mice were not significantly different to CIA controls, in terms..." add OIT before symptomatic and later in the sentence before asymptomatic.

Results - the asymptomatic group of OIT mice are not mentioned when it comes to numbers of Tregs in DLNs SF1

Results – “corroborating the clinical results, histopathological analysis of CIA joint...that was ameliorated in those OIT mice...” Ameliorated implies it has been made better, but you don’t know that there has been an improvement in these mice. Replace with “absent”.

Results section – can the authors clarify in Figure 3C are the measurements of IL17/IL22 in the joint from the “OIT” group from symptomatic, asymptomatic or both?

Results section - “Notably, despite the lower overall IL17 and IL22 producers in asymptomatic, but not groups or OIT mice.” Check for sense.

Results section “Oral administration of undenatured type II collagen protect against damage to the gut tissue in CIA”. The first sentence reads “Although inflammatory cytokines and cellular responses were reduced in the joint and DLN tissue of OIT animals...” This is now no longer true and should be removed. In the same section “This alteration was rectified in the small intestine of asymptomatic OIT mice...” this is only true of the duodenum and a trend in the ileum, but not the jejunum. Can this be modified.

Regarding IL22 producing cells in the MLN, ileum and colon. The authors say that there is a rewiring of IL22+ cell network in the gut in asymptomatic OIT mice. Can the authors additionally comment on the IL22 in their OIT symptomatic mice? Was this more similar to CIA mice or OIT asymptomatic.

Results – in reference to figure 7A the authors say “while asymptomatic OIT mice more resembled the profile of healthy tissue” in regards to UEA binding. This feels like an overstatement as the mean intensity is reduced , but just not significantly.

Figure 7 could go into supplementary data as this doesn’t contribute heavily to the main story.

I have also looked at the replies to Reviewer 3, as requested by editorial team and you can find my comments below the reviewer comments.

RA is a chronic, autoimmune disease affecting the synovial joints. In this paper, the authors used a model of Oral Immune Therapy (OIT), and treated CIA mice with undenatured collagen. They reported that the alleviated effect of undenatured collagen was due to the regulation of IL-17 and IL-22 cytokines in the joint and rewiring of associated gd T cell and ILC cytokine networks in the gut. This study is interesting, however, there are several major concerns for data interpretation and conclusion.

1. It is necessary to show the data in bar graph about the effect of OIT on CD4+ T cell number or frequency besides the radar charts in Fig 3G.

Okay, this data has been added.

2. In this study, authors indicate the OIT therapy may exert the benefit by regulating IL-17 and IL-22, they also showed the several cell types contributing to IL-17 and IL-22. Whether intestinal IL-22 is protective or pathogenic is still unclear, therefore, the conclusion is not convinced.

There is now significant discussion of the role of IL22 in both the gut and joint. I agree that a conclusion cannot be made from the data generated in this study, but that would be beyond the scope of this paper. The authors discussion encompasses the need to further explore this.

3. Authors showed that OIT can also alter gut glycosylation. Although in healthy gut, the support of IL-

22 on homeostatic expression of mucins is associated with fucosylation, it is unclear whether fucosylation is necessary in maintain gut integrity and in CIA alleviation.

I think the authors have addressed this point adequately.

4. Authors investigated the gut microbiota composition, while it is better to explain and interpret these data since there are many papers showing the GM with gut integrity or immune cells regulation in the gut. Since authors indicated the immune cells change in the gut, gut integrity, they should also discuss these relationship in the paper.

The results and discussion do include significant details around cross talk between the immune system and microbiota. I could not see any discussion of how the observed changes in the microbiota may impact gut integrity (or vice versa) to fully address this point, the authors could add some commentary of this.

5. Treg is usually detected as CD3+CD4+CD25+Foxp3+ in mice, what is the meaning to detect CD39+ or CD73+ Treg cells in this paper?

The authors have actually used CD3+ CD25+ and FoxP3+ to gate for Tregs, and have additionally looked within this subset for CD39+ and CD73+ Tregs. Their explanation for examining CD39 and CD73 is included in the results. I did note that on Supplementary Figure 1, in the gating strategy, the arrow is misplaced. Currently an arrow goes from the CD8+ population to the FoxP3/CD25 gating, this should be from CD4+ population.

Reviewer #2 (Remarks to the Author):

My concerns have been satisfactorily addressed by the authors.

Reviewer #1 (Remarks to the Author):

I am pleased to see that the authors have considered the difference between asymptomatic and symptomatic OIT mice, and I am happy that this has helped to draw conclusions about the differential effects on disease incidence and disease severity. I appreciate this was a significant amend to the presentation of the results. The manuscript now describes how OIT had a protective effect, and reduced the incidence of CIA. However, those OIT mice which did go on to develop symptoms exhibited the same degree of severity. Interestingly, the OIT prevented damage to the gut even in mice which went on to develop symptoms. To me this would suggest that breach of tolerance at mucosal surfaces does not have to occur in order to initiate joint disease and would help with the cause and effect argument. The findings have been in the main clearly communicated in the updated version. For absolute clarity, could the authors add into the beginning of the results section that all (I think it was 100%) the mice in the CIA group developed symptoms. This would be helpful. There are a few other points in the text (described below) where the differences between experimental groups need explaining.

We have indicated at the beginning of results that all mice in the CIA mice developed symptoms.

There is the implication that OIT somewhat reduced inflammatory responses in the joints of symptomatic mice – this seems to mostly hinge on data presented in Figure 3C/D. This figure needs to be clear on whether the OIT group refers to asymptomatic or symptomatic animals as this is a critical point.

Figure 3C shows data from asymptomatic mice only, this has been indicated in the text and in the figure. Figure 3D contains data from PMA/lo ex vivo stimulated lymph node cells, containing data from both symptomatic and asymptomatic mice. We have now clarified that this in the text.

The 16S rRNA seq adds further detail to the paper. There seems to be a striking reduction in diversity in the ileum of the asymptomatic OIT group in particular in the Shannon index [Shannon is misspelt in the figure itself and text] and the results section describes “lower diversity”. Yet this contrasts with what is said later on “Overall, OIT increased the diversity of bacteria in the ileum, ...” Can this be modified?

This has been corrected. That section was referring to the original Figure 8 about data in the colon, not ileum. This has been corrected to: ‘Compared to symptomatic mice, the asymptomatic OIT group increased the diversity of bacteria in the colon’.

Additionally, the PICRUSt2 analysis may need some further explaining. The pathways on the figure (SF7) are just labelled with the number. To say that OIT mice show a “distinct metabolite profile in the colon” is a bit misleading if this is an interpretation of figure S7B.

We have added some additional explanation. We have also included a reference for the PICRUSt2 package [46:Douglas, G. M. et al. PICRUSt2 for prediction of

metagenome functions. Nat Biotechnol 38, 685-688, doi:10.1038/s41587-020-0548-6 (2020)].

We agree that the interpretation of supplemental figure 7B could be misleading. We have revised that paragraph, and we have added the pathways names to allow a more comprehensive analysis and interpretation of the data.

The revised section in the results regarding this is:

'To gain some insight into the potential biological activities of microbial communities without directly measuring gene expression or protein function, we conducted some metagenomic function prediction based on the 16S sequencing using the bioinformatic tool PICRUSt2 (46). This analysis indicated differences among naïve, CIA and OIT groups, particularly in the ileum of OIT asymptomatic mice (Supplementary Figure 7A), but also in the colon (Supplementary Figure 7B). Although further experimental work is required to demonstrate this, this data suggest that effective OIT could impact of the functional host-microbiome crosstalk during chronic arthritis'.

The authors have addressed all my other points and I thank them for these responses.

Additional comments

Abstract – “Incidence was reduced to 50%, correlating with down regulation of interleukin 17 and IL 22 in the joints” Does this statement still hold? Is it not referring to the previous version of the data?

We have modified the abstract to reflect the new analysis, separating symptomatic and asymptomatic mice:

'OIT reduced disease incidence to 50%, with reduced expression of IL-17 and IL-22 in the joints of asymptomatic mice. Moreover, whilst the gut tissue of arthritic mice shows substantial damage and activation of tissue-specific immune networks, oral administration of undenatured type II collagen protects against gut pathology in all mice, both symptomatic and asymptomatic, rewiring IL-17/IL-22 networks'

Introduction “Current immunosuppressive treatment still shows 20-30% of non-responders” check for sense?

We have modified this sentence: ‘Despite advancements in immunosuppressive treatments, approximately 20-30% of patients still do not respond to them’

Introduction “Mechanistically, oral collagen administration promotes regulatory T cell function...” where? Can the authors state in the gut, joint or both?

These studies have been conducted in the gut tissue. This has been indicated in the revised manuscript.

Results – When the prophylactic undenatured type II collagen oral immunotherapy regime is first mentioned (line 5 of the results section) can the authors briefly explain that this starts 2 weeks prior to CIA induction and is 3x per week – for clarity.

This has been added.

‘OIT started 2 weeks prior to CIA induction, and oral gavage of undenatured type II collagen was applied 3 times per week’.

Results – “To evaluate whether OIT...to measure paw swelling and relatively weight change” should be relative not relatively.

This has been corrected.

Results – “symptomatic mice were not significantly different to CIA controls, in terms...” add OIT before symptomatic and later in the sentence before asymptomatic.

This has been added.

Results - the asymptomatic group of OIT mice are not mentioned when it comes to numbers of Tregs in DLNs SF1

We think that these are mentioned. In relation with the data shown in SF1 we have:

‘we evaluated the levels of CD3+CD25+FoxP3+ Tregs in the DLNs of asymptomatic OIT mice compared to those of the naïve and CIA groups (Supplementary Figure 1)’ and later on ‘We did not observe any expansion of Tregs in asymptomatic OIT mice at the cull day (day 33)’.

Results – “corroborating the clinical results, histopathological analysis of CIA joint...that was ameliorated in those OIT mice...” Ameliorated implies it has been made better, but you don’t know that there has been an improvement in these mice. Replace with “absent”.

Thanks, that is a good point. We have replaced *ameliorated* with *absent*.

Results section – can the authors clarify in Figure 3C are the measurements of IL17/IL22 in the joint from the “OIT” group from symptomatic, asymptomatic or both?

This has been changed, indicating that Figure 3C shows asymptomatic mice.

Results section - “Notably, despite the lower overall IL17 and IL22 producers in asymptomatic, but not groups or OIT mice.” Check for sense.

Apologies, we overlooked this mistake, this should not be there. We have deleted it in the revised manuscript.

Results section “Oral administration of undenatured type II collagen protect against damage to the gut tissue in CIA”. The first sentence reads “Although inflammatory

cytokines and cellular responses were reduced in the joint and DLN tissue of OIT animals...” This is now no longer true and should be removed. In the same section “This alteration was rectified in the small intestine of asymptomatic OIT mice...” this is only true of the duodenum and a trend in the ileum, but not the jejunum. Can this be modified.

These have been removed and replaced as suggested.

‘This alteration was rectified in the duodenum of asymptomatic OIT mice, with a similar trend in jejunum and ileum. This effect was not observed in the colon’.

We have stated that this is not observed in the colon, since OIT asymptomatic and naïve groups are significantly different. We say that the jejunum and ileum keep the same trend than the duodenum because Naïve and CIA groups are significantly different in both cases, but Naïve and OIT asymptomatic are not. In the colon, both CIA and OIT asymptomatic are significantly different to Naïve mice.

Regarding IL22 producing cells in the MLN, ileum and colon. The authors say that there is a rewiring of IL22+ cell network in the gut in asymptomatic OIT mice. Can the authors additionally comment on the IL22 in their OIT symptomatic mice? Was this more similar to CIA mice or OIT asymptomatic.

We cannot comment on the symptomatic mice because we don't have these data for the symptomatic OIT mice, at least not confidently. We have some preliminary data that we are analysing, but we can't provide any conclusive answer. To study immune networks in the MLN, ileum, and colon, we worked with the asymptomatic OIT mice, with the main goal of investigating the local immunological pathways responsible for protection, as we stated in the results section. The protocol to isolate single cells from gut tissue is lengthy and contains critical steps to digest the tissue that are highly sensitive to overexposure. The data shown in the manuscript required 32 hours of continuous work, but more importantly, when we tried a larger number of animals, we realized that the difference in digestion time between the first and the last sample was too high, compromising the experiment. Now that we have a better understanding of the effect of OIT in symptomatic and asymptomatic mice, our future studies will aim to design more targeted experiments to investigate specific pathways modulated in the MLN and gut tissue of both symptomatic and asymptomatic OIT mice.

Results – in reference to figure 7A the authors say “while asymptomatic OIT mice more resembled the profile of healthy tissue” in regards to UEA binding. This feels like an overstatement as the mean intensity is reduced , but just not significantly.

We understand this point, but we would think that the sentence aligns with the data. In the Ileum, the CIA group has a significant reduction in UEA binding, something that does not happen between Naïve and OIT mice. This indicates that Naïve and OIT groups are not different, in other words, OIT reverts the effect observed in CIA.

We have modified the paragraph for clarity:

'terminal fucosylation (UEA binding; terminal alpha[1,2] linked fucose residues) was significantly reduced in the ileum of CIA mice, while asymptomatic OIT mice more resembled the profile of healthy tissue, as they were not significantly different to the Naïve group (Figure 7A)'

Figure 7 could go into supplementary data as this doesn't contribute heavily to the main story.

We have thought about this, and we consider these results relevant. We would prefer to keep them in the main manuscript because we want to provide a global view of the mechanisms. Changes in the gut glycosylation are directly connected to changes in the microbiome composition, which can be related to the activity of immune networks. In our work, we show an integrative approach, showing a correlation between these factors. Future studies in our group will aim to address potential causality, but at this stage we think that this potential link could lose strength if the data were moved to supplementary. However, we also acknowledge that the manuscript will be more focused if figure 7 was supplementary, and we could change that if the reviewer considers this essential.

I have also looked at the replies to Reviewer 3, as requested by editorial team and you can find my comments below the reviewer comments.

We would like to thank reviewer 1 for their constructive feedback and their effort to go through additional comments. We have checked these as well, moving the misplaced arrow in SF1.

RA is a chronic, autoimmune disease affecting the synovial joints. In this paper, the authors used a model of Oral Immune Therapy (OIT), and treated CIA mice with undenatured collagen. They reported that the alleviated effect of undenatured collagen was due to the regulation of IL-17 and IL-22 cytokines in the joint and rewiring of associated gd T cell and ILC cytokine networks in the gut. This study is interesting, however, there are several major concerns for data interpretation and conclusion.

1. It is necessary to show the data in bar graph about the effect of OIT on CD4+ T cell number or frequency besides the radar charts in Fig 3G.

Okay, this data has been added.

2. In this study, authors indicate the OIT therapy may exert the benefit by regulating IL-17 and IL-22, they also showed the several cell types contributing to IL-17 and IL-22. Whether intestinal IL-22 is protective or pathogenic is still unclear, therefore, the conclusion is not convinced.

There is now significant discussion of the role of IL22 in both the gut and joint. I agree that a conclusion cannot be made from the data generated in this study, but that would be beyond the scope of this paper. The authors discussion encompasses the need to further explore this.

3. Authors showed that OIT can also alter gut glycosylation. Although in healthy gut, the support of IL-22 on homeostatic expression of mucins is associated with fucosylation, it is unclear whether fucosylation is necessary to maintain gut integrity and in CIA alleviation.

I think the authors have addressed this point adequately.

4. Authors investigated the gut microbiota composition, while it is better to explain and interpret these data since there are many papers showing the GM with gut integrity or immune cells regulation in the gut. Since authors indicated the immune cells change in the gut, gut integrity, they should also discuss these relationships in the paper.

The results and discussion do include significant details around cross talk between the immune system and microbiota. I could not see any discussion of how the observed changes in the microbiota may impact gut integrity (or vice versa) to fully address this point, the authors could add some commentary of this.

5. Treg is usually detected as CD3⁺CD4⁺CD25⁺Foxp3⁺ in mice, what is the meaning to detect CD39⁺ or CD73⁺ Treg cells in this paper?

The authors have actually used CD3⁺ CD25⁺ and FoxP3⁺ to gate for Tregs, and have additionally looked within this subset for CD39⁺ and CD73⁺ Tregs. Their explanation for examining CD39 and CD73 is included in the results. I did note that on Supplementary Figure 1, in the gating strategy, the arrow is misplaced. Currently an arrow goes from the CD8⁺ population to the FoxP3/CD25 gating, this should be from CD4⁺ population.

REVIEWERS' COMMENTS:

Reviewer #1 (Remarks to the Author):

I am happy with the authors' responses to my comments.